# Continuous-Time Analysis of Adaptive Optimization and Normalization

## Abstract

Adaptive optimization algorithms, particularly Adam and its variant AdamW, are fundamental components of modern deep learning. However, their training dynamics lack comprehensive theoretical understanding, with limited insight into why common practices—such as specific hyperparameter choices and normalization layers—contribute to successful generalization. This work presents a continuous-time formulation of Adam and AdamW, facilitating a tractable analysis of training dynamics that can shed light on such practical questions. We theoretically derive a stable region for Adam's hyperparameters $(\beta, \gamma)$ that ensures bounded updates, empirically verifying these predictions by observing unstable exponential growth of parameter updates outside this region. Furthermore, we theoretically justify the success of normalization layers by uncovering an implicit meta-adaptive effect of scale-invariant architectural components. This insight leads to an explicit optimizer, 2-Adam, which we generalize to $k$-Adam—an optimizer that applies an adaptive normalization procedure $k$ times, encompassing Adam (corresponding to $k = 1$) and Adam with a normalization layer (corresponding to $k = 2$). Overall, our continuous-time formulation of Adam facilitates a principled analysis, offering deeper understanding of optimal hyperparameter choices and architectural decisions in modern deep learning.

## 1 Introduction

Adaptive optimization algorithms have become an essential component of modern deep learning, providing significant benefits to the training of neural networks compared to their non-adaptive counterparts. Among these algorithms, Adam (Kingma & Ba, 2017) and its variant AdamW (Loshchilov & Hutter, 2019) have become widely used in practice, featuring both an adaptive learning rate (in the sense of RMSprop (Tieleman & Hinton, 2012)) and an adaptive gradient direction (in the sense of momentum (Polyak, 1964)). Despite their wide success, a theoretical understanding of their training dynamics is lacking. Previous theoretical work on adaptive optimization mostly focuses on results regarding asymptotic convergence rates under specific conditions (Chen et al., 2019; Li & Orabona, 2019; Zhou et al., 2024; Barakat & Bianchi, 2020; da Silva & Gazeau, 2019), however, there is limited theoretical insight into why common practices – such as typical hyperparameter choices, and the use of normalization layers (e.g. layer-norm) – contribute towards successful generalization.

In this work, we demonstrate that continuous-time models of optimization can shed light on such questions. A continuous-time approach allows for a mathematically tractable analysis that can leverage the tools of calculus – namely, differential equations. Past work has utilized continuous-time models in contexts of practical interest (Tanaka & Kunin, 2021; Kunin et al., 2021; Zhao et al., 2023; Chen et al., 2024a; Elkabetz & Cohen, 2021), though such work does not consider optimizers with adaptive learning rates (e.g. Adam), hence practical insight into modern deep learning is limited.

Our contributions can be summarized as:

**Sec. 3. Theory of adaptive hyperparameters.** We determine a theoretical stability region for Adam's adaptive hyperparameters $(\beta, \gamma)$ that ensures bounded parameter updates and stable training, which we verify empirically. This result implies that instability may occur when the hyperparameters move outside the stable region, a phenomenon we empirically verify,

exhibiting *predictable* exponential growth. We observe a faster rate of generalization when $(\beta, \gamma)$ is further from the instability boundary.

**Sec. 4. Implicit effect of scale invariance.** We perform a theoretical analysis of how scale invariant architectural components (e.g. layer-norm) influence Adam's learning dynamics, finding scale invariance to induce a *meta-adaptive* normalization effect. We convert this implicit effect to an *explicit* optimizer, which we name 2-Adam, and consider its extension $k$-Adam: a generalization of Adam/AdamW (which corresponds to $k = 1$) that performs a normalization procedure $k$ times successively.

We discuss related work further in Section 5.

## 2 CONTINUOUS-TIME FORMULATION OF ADAM

In this section we present a continuous-time formulation of the Adam (and AdamW) optimizer which forms the theoretical foundation for the rest of the paper. We derive a continuous-time expression for the Adam gradient update (utilized in Section 3 to derive a condition for bounded updates) and describe Adam as a second-order differential equation (used in Section 4 to interpret the implicit effect of scale invariance and motivate the $k$-Adam optimizer). Experimental details are left to Appendix H.

**Notation.** We write $||x|| := \sqrt{\sum_i x_i^2}$ and $||x||_\infty := \max_i |x_i|$ for $x \in \mathbb{R}^p$, and denote the inner product $\langle x, y \rangle \equiv \sum_i x_i y_i$. We denote elementwise squaring by $x^{\odot 2}$. We consider a loss function $L : \Theta \to \mathbb{R}$ where $\Theta \subseteq \mathbb{R}^p$, and $\theta_n \in \Theta$ denotes a model's parameters after $n$ discrete updates, with corresponding gradient $g_n := \nabla_\theta L(\theta_n)$, and initial parameter $\theta_0$. We will write $g_{0:n} \equiv (g_0, \ldots, g_n)$.

### 2.1 ADAM AND ADAPTIVE NORMALIZATION

We will first briefly define Adam and the concept of adaptive normalization. We also provide a brief summary of relevant adaptive optimizers (momentum and RMSprop) in Appendix A.

**Definition 1** (Adam). *Neglecting weight decay (i.e. $\lambda = 0$), Adam (and AdamW) possess the discrete-time update rule*

$$\theta_{n+1} = \theta_n - \eta u_n, \quad \text{where} \quad u_n := \frac{\sqrt{1 - \beta^{n+1}}}{1 - \gamma^{n+1}} \frac{m_n}{\sqrt{v_n}} \tag{1}$$

*for $n = 0, 1, 2, \ldots$, for learning rate $\eta \in \mathbb{R}_+$ and adaptive hyperparameters $(\beta, \gamma) \in [0, 1]^2$ with corresponding moving-averages*

$$m_n := \gamma m_{n-1} + (1 - \gamma) g_n, \quad v_n := \beta v_{n-1} + (1 - \beta) g_n^{\odot 2}$$

*where $(m_{-1}, v_{-1}) := (0, 0)$.*

The following notation will be useful in Section 4 for a clear description of our findings.

**Definition 2** (Adaptive normalization). *We say that $u_n$ is an adaptive normalization of the gradient history $g_{0:n}$ and equivalently write $u_n = \mathcal{A}_{\gamma, \beta}(g_{0:n})$, where for any sequence of tensors $x_{0:n} = (x_0, \ldots, x_n)$ we define*

$$\mathcal{A}_{\gamma, \beta}(x_{0:n}) := \frac{M_\gamma(x_{0:n})}{\sqrt{M_\beta(x_{0:n}^{\odot 2})}}$$

*defining the (bias-corrected) exponential moving-average*

$$M_\gamma(x_{0:n}) := \frac{1 - \gamma}{1 - \gamma^{n+1}} (x_n + \gamma x_{n-1} + \cdots + \gamma^n x_0)$$

### 2.2 PARAMETER DYNAMICS IN CONTINUOUS-TIME

We will work in continuous-time at a timescale of $\eta^p$ with discrete timesteps $t_n := n\eta^p$ (for some $p \geq 1$), writing e.g. $m_n = m(t_n)$. From here, we can derive leading order continuous-time expressions for the moving averages $m_n$ and $v_n$, as described by the following result:

**Lemma 1** (Appendix C). *Up to order $\eta^p$ and for $\beta, \gamma \in (0, 1)$, the continuous-time moving averages $m(t)$ and $v(t)$ satisfy the first-order differential equations*

$$\eta^p \gamma \dot{m}(t) + (1 - \gamma)m(t) = (1 - \gamma)g(t), \quad \eta^p \beta \dot{v}(t) + (1 - \beta)v(t) = (1 - \beta)g(t)^{\odot 2}$$

*with solutions*

$$m(t) = \int_0^t d\tau \; K_\gamma(\tau, t)g(\tau), \quad v(t) = \int_0^t d\tau \; K_\beta(\tau, t)g(\tau)^{\odot 2}, \tag{2}$$

*where $g(t) := \nabla_\theta L(\theta(t))$ and defining*

$$K_\alpha(\tau, t) := \frac{1 - \alpha}{\eta^p \alpha} \exp\left(-\frac{1 - \alpha}{\eta^p \alpha}(t - \tau)\right)$$

We neglect terms of order higher than $\eta^p$ as we implicitly consider the continuous-time limit of small/infinitesimal learning rate $\eta \to 0$, similarly to previous works (Malladi et al., 2024; da Silva & Gazeau, 2019; Barakat & Bianchi, 2020). In Figure 1 we provide empirical verification of this assumption for a learning rate of $10^{-3}$ (the PyTorch default learning rate for Adam). We note that an equivalent continuous-time derivation of a first-order expression for $v(t)$ has been derived previously in the context of adaptive optimization (Wang et al., 2021) (specifically, see Appendix A.3). An immediate consequence of Lemma 1 is a continuous-time expression for Adam's parameter update $u_n$:

**Proposition 1.** *Using the (order $\eta^p$) expressions for $m(t)$ and $v(t)$ from Lemma 1, we can write the continuous-time parameter update $u(t)$ as*

$$u(t) = \frac{\sqrt{1 - \beta^{1 + t/\eta^p}}}{1 - \gamma^{1 + t/\eta^p}} \frac{\int_0^t d\tau \; K_\gamma(\tau, t)g(\tau)}{\sqrt{\int_0^t d\tau \; K_\beta(\tau, t)g(\tau)^{\odot 2}}} \tag{3}$$

As we will see, Proposition 1 is central to the analysis of Section 3, allowing us to derive theoretical guarantees regarding training stability with respect to adaptive hyperparameters $(\beta, \gamma)$. Using this continuous-time expression for $u(t)$, we can model the continuous-time dynamics of $\theta(t)$:

**Proposition 2** (Appendix C). *Up to order $\eta^p$ and at timescale $p = 1$, AdamW with weight decay $\lambda$ is described by the second-order differential equation*

$$\lambda\theta(t) + \dot{\theta}(t) + \frac{1}{2}\eta\ddot{\theta}(t) = -u(t) \tag{4}$$

*with $u(t)$ as in Proposition 1.*

We can numerically solve Equation (4) to obtain a continuous-time parameter trajectory $\theta(t)$, with our method described explicitly in Appendix I. In Figure 1 we demonstrate that this continuous-time trajectory closely agrees with the true discrete-time trajectory. In Section 4 we will make use of Proposition 2 to interpret the implicit effect of scale invariance.

Though our above framework does not account for stochastic gradients, we note that throughout Section 3 and Section 4, our experimental setups make use of stochastic/mini-batched implementations of Adam, which we find to closely agree with our theoretical results. We describe how the above can be easily extended to include weight decay in Appendix C.

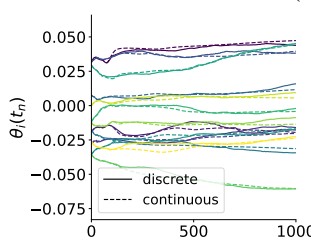

Figure 1: **Continuous-time model closely agrees with discrete-time trajectories.** We plot the discrete-time and continuous-time trajectories for 16 randomly chosen parameters from a transformer model.

## 3 THEORY OF ADAPTIVE HYPERPARAMETER CHOICE

We will now begin to apply the continuous-time framework presented in the previous section to understanding aspects of adaptive optimization from a theoretical perspective. In this section we will present a theory-driven account of adaptive hyperparameter choice for Adam, understanding the values of $(\beta, \gamma)$ that result in stable training and effective generalization.

First we will use the continuous-time expression for Adam's update (Equation (3)) to derive a theoretical region of hyperparameter space $\mathcal{B}_+$ for which updates are provably bounded (Section 3.1). We will then empirically verify that this region indeed exhibits stable training, with unstable training in the complementary region $\mathcal{B}_-$ (exhibiting a *predictable* exponential growth) (Section 3.2), and observe how generalization performance varies in the regions $\mathcal{B}_+$ and $\mathcal{B}_-$ (Section 3.3).

## 3.1 DERIVING A CONDITION FOR BOUNDED UPDATES

The continuous-time expression for Adam's update as in Proposition 1 has an immediate consequence in regards to bounding the *max-update* $||u_n||_\infty \equiv ||\theta_{n+1} - \theta_n||_\infty / \eta$. The max-update quantifies the maximal parameter change (across *all* parameters) at a given step, hence upper bounds on this quantity also hold identically for the parameter change of any arbitrary parameter. We have the result:

**Theorem 1** (Appendix D). *Using the order $\eta^p$ expression for $u_n = u(t_n)$ from Proposition 1 and assuming $\beta, \gamma \in (0, 1)$, the max-update can be bounded as*

$$||u_n||_\infty \leq \frac{\sqrt{1-\beta^{n+1}}}{1-\gamma^{n+1}} \frac{1-\gamma}{\gamma} \sqrt{\frac{\beta}{1-\beta}} B_n(\beta, \gamma) \tag{5}$$

*where*

$$B_n(\beta, \gamma) := \begin{cases} 1/\sqrt{C(\beta,\gamma)}, & C(\beta,\gamma) > 0, \\ \sqrt{n}, & C(\beta,\gamma) = 0, \\ \exp(n|C(\beta,\gamma)|/2)/\sqrt{|C(\beta,\gamma)|}, & C(\beta,\gamma) < 0 \end{cases}$$

*defining* $C(\beta, \gamma) := (2\beta(1-\gamma) - \gamma(1-\beta))/\beta\gamma$.

We highlight that this bound is only possible because of the specific form of Adam's update $u_n$: a moving-average of $g_{0:n}$ divided by the square-root of a moving-average of $g_{0:n}^{\odot 2}$, which allows us to apply the Cauchy-Schwarz inequality. Analogous bounds in a discrete-time context have been similarly derived in previous works (Zou et al., 2021; Zhang et al., 2024; Xie & Li, 2024; Hong & Lin, 2024), also making use of the Cauchy-Schwarz inequality (as we do in Appendix D). We note that our proof of Theorem 1 relies on assuming non-stochastic gradients, however, we will see further below that this result makes accurate predictions in stochastic empirical contexts. Further, this bound is independent of the chosen timescale $p$. The special cases of RMSprop and signSGD are discussed in Appendix D.

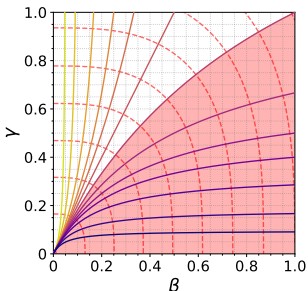

Figure 2: Visualization of level curves $\mathcal{B}_c$ (solid lines) and normal curves $\mathcal{C}_{\beta,\gamma}$ (dashed red lines). Level curves are coloured based on their value of $C(\beta, \gamma)$, (i.e. purple has most positive value, yellow most negative). The bounded-update region $\mathcal{B}_+$ is highlighted in red.

We define the bounded-update region $\mathcal{B}_+ := \{(\beta, \gamma) : C(\beta, \gamma) > 0\}$, and the complementary region $\mathcal{B}_- := \{(\beta, \gamma) : C(\beta, \gamma) < 0\}$. From Equation (5), we can bound the max-update by a constant independent of $n$ when $(\beta, \gamma) \in \mathcal{B}_+$. Outside of $\mathcal{B}_+$, the bound depends on $n$ and diverges over training as $n \to \infty$. This is suggestive that we may observe stable training when $(\beta, \gamma) \in \mathcal{B}_+$, and that the max-update may grow exponentially (at a rate/exponent proportional to $|C(\beta, \gamma)|$) when $(\beta, \gamma) \in \mathcal{B}_-$. Indeed, we will verify this phenomena empirically in Section 3.2. We comment on the case $(\beta, \gamma) \in \mathcal{B}_0$ in Appendix E. It is easy to show that $\beta > \gamma$ is a sufficient condition for $(\beta, \gamma) \in \mathcal{B}_+$, meaning that the choice of hyperparameters typically chosen in practice – e.g. the PyTorch default values $(\tilde{\beta}, \tilde{\gamma}) := (0.999, 0.9)$ – lie within $\mathcal{B}_+$.

For the following, we define the *level curves* $\mathcal{B}_c := \{(\beta, \gamma) : C(\beta, \gamma) = c\}$, and consider *normal curves* perpendicular to these level curves, visualized in Figure 2. We will denote the (unique) normal curve passing through the point $(\beta, \gamma)$ by $\mathcal{C}_{\beta,\gamma}$. In Section 3.2 and Section 3.3 we will analyze how $C(\beta, \gamma)$ correlates with training stability & generalization performance by observing how such properties change as we move along the normal curve $\mathcal{C}_{\tilde{\beta},\tilde{\gamma}}$ in hyperparameter space, since $C(\beta, \gamma)$ varies monotonically along a normal curve (by construction).

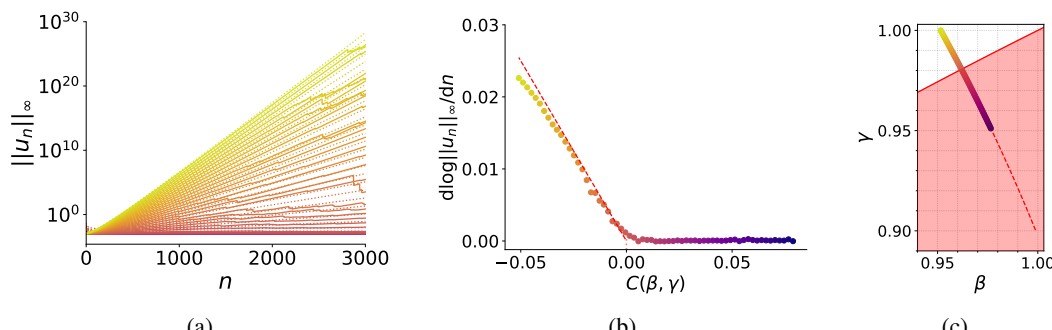

(a)                 (b)                 (c)

Figure 3: **Max-update bound accurately predicts stable region and unstable exponent of divergence.** For a range of adaptive hyperparameter values $(\beta, \gamma)$, we plot (a) the max-update $||u_n||_\infty \equiv ||\theta_n - \theta_{n-1}||_\infty / \eta$ over training iterations $n$, and (b) the slope $d \log ||u_n||_\infty / dn$ of the log-max-update at iteration $n = 1000$ (in order to interpret exponential growth). In (a) we visualize the bounds of Equation (5) as dotted lines, and in (b) we denote the predicted slope/exponent $|C(\beta, \gamma)|/2$ (when $C(\beta, \gamma) < 0$) as a dashed line. We consider $64$ choices for $(\beta, \gamma)$, visualized in (c), taken uniformly along a section of the normal curve $\mathcal{C}_{\tilde{\beta}, \tilde{\gamma}}$ passing through the point $(\tilde{\beta}, \tilde{\gamma}) = (0.999, 0.9)$.

### 3.2   EMPIRICAL ANALYSIS OF MAX-UPDATE

We now consider the empirical implications of Equation (5), assessing whether these bounds indeed hold in practice and whether $C(\beta, \gamma)$ is predictive of training stability. We train a decoder-only transformer model with 30 million parameters on the C4 dataset (details left to Appendix H) and observe how the max-update $||u_n||_\infty$ evolves over training iterations $n$. Specifically, in Figure 3, we consider the normal curve $\mathcal{C}_{\tilde{\beta}, \tilde{\gamma}}$ passing through the typical hyperparameter values $(\tilde{\beta}, \tilde{\gamma}) = (0.999, 0.9)$, taking 64 points $(\beta, \gamma)$ uniformly along this curve (which we visualize in Figure 3c). For each point $(\beta, \gamma)$, we plot the max-update trajectory (Figure 3a), finding that the theoretical bounds of Equation (5) are satisfied in both regions $\mathcal{B}_+$ and $\mathcal{B}_-$, observing well-behaved bounded growth in $\mathcal{B}_+$, whereas $\mathcal{B}_-$ exhibits exponential growth at a rate that appears correlated with $|C(\beta, \gamma)|$ as predicted by Equation (5). We more closely verify this phenomena in Figure 3b, finding that in $\mathcal{B}_-$, the slope $d \log ||u_n||_\infty / dn$ has a near-perfect agreement with the theoretically predicted growth rate of $|C(\beta, \gamma)|/2$. It is surprising that not only are the theoretical bounds (Equation (5)) satisfied in practice, but the bounds are very accurate models of the true empirical dynamics, with exponential growth occurring almost immediately after entering $\mathcal{B}_-$. We look more closely at the results of Figure 3a in the case of $(\beta, \gamma) \in \mathcal{B}_+$ in Appendix L. We comment on correcting the exponential growth in $\mathcal{B}_-$ via learning rate annealing in Appendix N.

These results partly justify the success of typical values $(\tilde{\beta}, \tilde{\gamma}) = (0.999, 0.9)$ in practice: these values lie in $\mathcal{B}_+$ and hence benefit from theoretical guarantees regarding training stability (which, as we have seen, are faithful to practice). Can we obtain a more fine-grained picture as to what choices of $(\beta, \gamma)$ within the region $\mathcal{B}_+$ will result in successful generalization? We will now explore this.

### 3.3   PROPERTIES OF GENERALIZATION IN $\mathcal{B}_+$

We will now assess how generalization performance varies in $\mathcal{B}_+$, and particularly, how the value of $C(\beta, \gamma)$ correlates with generalization performance. In Figure 4 we observe how test loss varies along the normal curve $\mathcal{C}_{\tilde{\beta}, \tilde{\gamma}}$. taking 128 points uniformly along the entire curve (visualized in Figure 4c). We see that larger values of $C(\beta, \gamma)$ display faster generalization, as seen in Figure 4a and highlighted at iteration 1000 in Figure 4b. After sufficient training, i.e. at iteration 3000 in Figure 4b, all points in $\mathcal{B}_+$ achieve roughly the same test loss. The test loss exhibits a rapid increase after entering the region $\mathcal{B}_-$, which is to be expected given the unstable exponential growth in the max-update shown in Section 3.2.

These results suggest that for a given normal curve, the point $(\beta, \gamma)$ with the largest value of $C(\beta, \gamma)$ generalizes the fastest along the curve. We provide supporting evidence for this claim, finding similar results for the normal curve that instead passes through the hyperparameter values $(0.95, 0.9)$, in

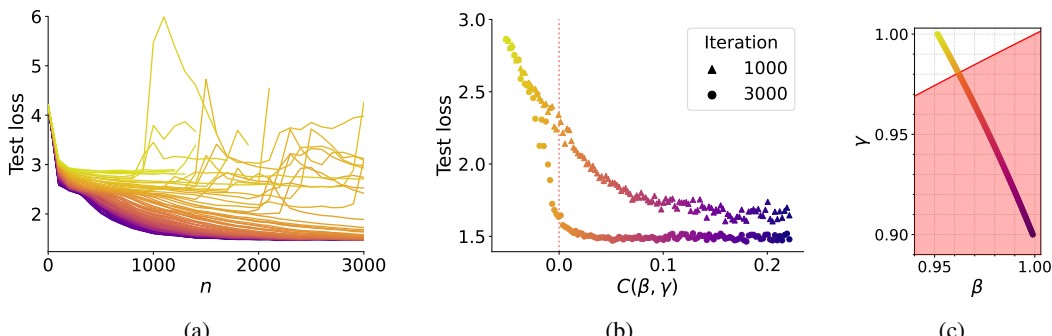

(a)  (b)  (c)

Figure 4: **Our theory accurately predicts the divergence of test loss across Adam's hyperparameter space.** For a range of values $(\beta, \gamma)$, we plot (a) the test loss over training iterations $n$, and (b) the best test loss achieved over the first 1000 and 3000 iterations. We consider 128 choices for $(\beta, \gamma)$, visualized in (c), taken uniformly along the entire normal curve $\mathcal{C}_{\tilde{\beta}, \tilde{\gamma}}$. The rightmost point in (c) corresponds to $(\tilde{\beta}, \tilde{\gamma}) = (0.999, 0.9)$.

Appendix M. This would further justify why the values $(\tilde{\beta}, \tilde{\gamma})$ succeed in practice (since $\tilde{\beta} \approx 1$, hence $C(\tilde{\beta}, \tilde{\gamma})$ is large relative to other points along its normal curve), however unlike the results of Section 3.2, this claim lacks an explicit supporting theoretical result; it is only suggestive from Equation (5) that a larger value of $C(\beta, \gamma)$ may correlate with better training properties. We leave direct theoretical guarantees regarding the rate of generalization to future work.

## 4  IMPLICIT ADAPTIVE ROLE OF SCALE INVARIANCE

The presence of scale-invariant architectural components has become ubiquitous in modern deep learning. One particular instance of such components are normalization layers, such as layer-norm (Ba et al., 2016), batch-norm (Ioffe & Szegedy, 2015), and qk-norm (Dehghani et al., 2023; Gilmer et al., 2023). It has been observed that normalization layers provide an *implicit* beneficial effect to training, in comparison to explicitly fixing the norm of weights after each training step (Tanaka & Kunin, 2021; Lubana et al., 2021; Santurkar et al., 2019). Such benefits include smoothing of the loss landscape (Santurkar et al., 2019; Kohler et al., 2018; Karakida et al., 2019; Lyu et al., 2023) and prevention of rank collapse (Daneshmand et al., 2020; Dong et al., 2023; Noci et al., 2022). In this section, we will apply the continuous-time framework of Section 2 to better understand the implicit role of scale invariance, allowing us to gain an understanding as to how normalization layers contribute towards successful generalization.

In this section, we will first briefly discuss scale-invariant maps (Section 4.1) and then make use of the second-order differential equation for AdamW (Proposition 2) to solve for the dynamics of a scale invariant weight, uncovering an implicit *meta-adaptive* effect of scale invariance (Section 4.2). We then convert this implicit effect into an explicit optimizer, 2-Adam, and find preliminary evidence of 2-Adam outperforming Adam in image classification and language modeling settings (Section 4.3).

### 4.1  SCALE-INVARIANT MAPS

A function $f : \Theta \to \mathbb{R}$ is *scale invariant* with respect to a weight $W \in \mathcal{W} \subseteq \Theta$ if and only if $f(\theta)$ remains unchanged under the transformation $W \mapsto \alpha W$ for all $\alpha \in \mathbb{R}$. As a concrete example, qk-norm (Dehghani et al., 2023; Gilmer et al., 2023) applies a layer-norm to the query and key projections in a transformer, with the attention logits $z_{ij}$ of a particular attention head taking the form,

$$z_{ij} := \text{LN}(W_Q x_i)^T \text{LN}(W_K x_j)$$

for embeddings $x_i \in \mathbb{R}^d$, query matrix $W_Q \in \mathbb{R}^{d_h \times d}$, key matrix $W_K \in \mathbb{R}^{d_h \times d}$, and layer-norm $\text{LN}(\cdot)$. The resulting attention logits $z_{ij}$ are scale-invariant with respect to $W_Q$ and $W_K$, and hence the loss function $L$ is equivalently scale-invariant. We have the following result:

**Lemma 2** (Appendix K). *For a loss function $L : \Theta \to \mathbb{R}$ and weight $W \in \mathcal{W} \subseteq \Theta$, if $L$ is scale invariant with respect to $W$, then*

$$\langle W(t), g_W(t) \rangle = 0 \quad \forall \, t \tag{6}$$

*where $g_W(t) := \nabla_W L(\theta(t))$ is the continuous-time gradient associated with $W$.*

Lemma 2 will be utilized in the following to show that scale-invariant maps possess an implicit *meta-adaptive* effect.

## 4.2 UNCOVERING AN IMPLICIT META-ADAPTIVE EFFECT

To perform a theoretical analysis of how scale invariance influences training dynamics, we will make use of the second-order differential equation associated with AdamW at timescale $p = 1$ (Proposition 2). Note that the accuracy of this setup was verified in Figure 1. We will consider a scale invariant weight $W$, for example, a key/query matrix under qk-norm. From Proposition 2, we can describe the continuous-time dynamics of $W$ up to order $\eta$ by the differential equation

$$\frac{1}{2}\eta \ddot{W}(t) + \dot{W}(t) + \lambda W(t) = -u_W(t) \tag{7}$$

with $u_W$ defined analogously to $u$, replacing $g$ with $g_W$ in Equation (3). In order for a tractable theoretical analysis, we will make the following two assumptions:

**Assumption 1.** *We will neglect $\langle W(t), g_W(\tau) \rangle$ for $\tau \leq t$, treating as second-order in $(\eta, \lambda)$.*

**Assumption 2.** *Using the expression for $v_W(t)$ from Lemma 1, we will use a coarse-graining approximation for $v_W(t)$:*

$$v_W(t) \equiv \int_0^t d\tau \, K_\beta(\tau, t) g_W(\tau)^{\odot 2} \approx \int_0^t d\tau \, \tilde{K}_\beta(\tau, t) ||g_W(\tau)||^2 =: \tilde{v}_W(t)$$

*with $\tilde{K}_\beta(\tau, t) := K_\beta(\tau, t)/N$ where $N$ is the total number of entries of $W$.*

Assumption 1 is supported by the property of scale invariance described by Lemma 2, and indeed, in Appendix G we find this assumption to hold empirically, with $\langle W(t), g_W(\tau) \rangle$ for $\tau \leq t$ negligible in practice (and in the case of $W$ not being scale-invariant, we find it to be *non-negligible*). Assumption 2 can be interpreted as replacing all entries of $g_W(\tau)^{\odot 2}$ with the average entry across the tensor, $||g_W(\tau)||^2/N$, in the expression for $v_W(t)$, i.e. assuming no variability in adaptive scaling across different parameters. We also find empirical support for this assumption in Appendix G, finding that replacing $v_W$ with its coarse-grained version $\tilde{v}_W$ has little effect on training dynamics. Under these two assumptions and using Equation (7), we have the following result:

**Theorem 2** (Appendix J). *For a scale invariant weight $W \in \mathcal{W} \subseteq \Theta$ satisfying Equation (7), and under Assumption 1 and Assumption 2, we can write (up to order $\eta$)*

$$||W(t)||^2 = ||W(0)||^2 e^{-2\lambda t} + \eta \int_0^t d\tau \, e^{-2\lambda(t-\tau)} ||u_W(\tau)||^2 \tag{8}$$

We empirically verify that Equation (8) is accurate to the true discrete-time trajectory of $||W(t)||^2$ in Figure 5. Importantly, Equation (8) has an ***identical form*** to an exponential moving-average of $||u_W||^2$ in continuous-time (see Appendix J for details). Given that $W$ is scale invariant (i.e. loss function independent of $||W||$), it is most relevant to consider the dynamics of $\hat{W}$, the unit direction associated with $W \equiv ||W||\hat{W}$. From Equation (7) we have the following result:

**Proposition 3** (Appendix J). *Up to first order in $(\eta, \lambda)$, the dynamics of $\hat{W}(t)$ are governed by*

$$\frac{1}{2}\eta \ddot{\hat{W}}(t) + \dot{\hat{W}}(t) + \Lambda \hat{W}(t) = -\frac{1}{||W(t)||} u_W(t) \tag{9}$$

*defining $\Lambda := \frac{1}{2}\eta r ||\dot{\hat{W}}||^2$.*

We highlight that Equation (9) has the same form as Equation (7), with weight decay $\Lambda$ instead of $\lambda$, and updating by $u_W/||W||$ instead of $u_W$. Given that $||W||^2$ takes the form of a moving-average of

---

**Algorithm 1** $k$-Adam update rule

---

**given** learning rate $\eta$, weight decay $\lambda$, *coupled* $\in$ {False, True}, hyperparameters $(\beta_{1:k}, \gamma_{1:k})$, epsilon $\epsilon$ ($10^{-30}$ by default)
**initialize** step count $n \leftarrow 0$, initial parameter $\theta_0 \in \mathbb{R}^p$, $(m_i, v_i) \leftarrow (0, 0)$ for $i = 1, \ldots, k$
**repeat**
    $g \leftarrow \nabla_\theta L(\theta_n)$
    **if** *coupled* **then**
        $g \leftarrow g + \lambda\theta_n$
    $\hat{g} \leftarrow g$
    **for** $i = 1, \ldots, k$ **do**
        $m_i \leftarrow \gamma_i m_i + (1 - \gamma_i)\hat{g}$
        $v_i \leftarrow \beta_i v_i + (1 - \beta_i)\hat{g}^{\odot 2}$
        $C \leftarrow \sqrt{1 - \beta_i^{n+1}}/(1 - \gamma_i^{n+1})$
        $\hat{g} \leftarrow C m_i/(\sqrt{v_i} + \epsilon)$
    $u \leftarrow \hat{g}$
    **if** not *coupled* **then**
        $u \leftarrow u + \lambda\theta_n$
    $\theta_{n+1} \leftarrow \theta_n - \eta u$
    $n \leftarrow n + 1$
**return** parameter trajectory $(\theta_0, \theta_1, \theta_2, \ldots)$

---

$||u_W||^2$ (Equation (8)), this allows us to view $u_W/||W||$ as an ***adaptive scaling*** of $u_W$, equivalent to the effect of RMSprop (scaling by the square root of a moving-average). We can describe this effect equivalently using the notation of Section 2.1. Specifically, adaptive normalization refers to $u_n \equiv \mathcal{A}_{\gamma,\beta}(g_{0:n})$, and we will use the phrase ***meta-adaptive normalization*** to refer to the concept of **applying additional adaptive normalization $\mathcal{A}_{\gamma',\beta'}$ to an *already normalized* history of updates** $\boldsymbol{u_{0:n}}$, described by $u_n^{(2)} := \mathcal{A}_{\gamma',\beta'}(u_{0:n})$. From the results above, we have found that: **the direction $\hat{W}$ of a scale invariant weight updates by a meta-adaptive normalized update**, described by $\mathcal{A}_{\gamma',\beta'}(u_{0:n})$ for $\gamma' = 0$. We will now make this effect explicit with the 2-Adam optimizer.

### 4.3 Implicit effect as an explicit optimizer

The analysis of Section 4.2 has shown that scale invariance has an adaptive influence on the dynamics of Adam, equivalent to **applying an adaptive normalization procedure twice in succession**. We will now make this concept explicit with the $k$-Adam optimizer:

**Definition 3** ($k$-Adam)**.** *The $k$-Adam optimizer applies an adaptive normalization $k$ times in succession, possessing hyperparameters $(\beta_{1:k}, \gamma_{1:k})$ and update rule:*

$$k\text{-Adam:} \quad \theta_{n+1} = \theta_n - \eta u_n^{(k)}$$

*recursively defining*

$$u_n^{(i)} := \mathcal{A}_{\gamma_i,\beta_i}(u_{0:n}^{(i-1)}) \quad for \quad i = 1, \ldots, k$$

*using the notation of Section 2.1, and with $u_{0:n}^{(0)} \equiv g_{0:n}$.*

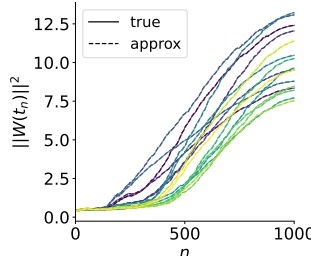

Figure 5: **Norm approximation agrees with true norm.** The true trajectory of $||W||^2$ compared to the approximation (dashed line) of Theorem 2 for 16 randomly chosen query/key matrices.

We present an explicit algorithm for implementing $k$-Adam in Algorithm 1. Note that 1-Adam is equivalent to Adam. $k$-Adam allows us to *explicitly* capture the implicit effect of scale invariance found in Section 4.2. More precisely, 2-Adam with $\gamma_2 = 0$ corresponds to the implicit dynamics of Adam with a normalization layer. As the implicit effects of scale invariance have been found to be beneficial in practice, we may expect 2-Adam to outperform Adam, which we indeed find supporting evidence of further below. We note that update bounds analogous to those shown for Adam (Theorem 1) hold identically for $k$-Adam, as shown by the following result:

**Theorem 3.** *Using order $\eta^p$ expressions for associated moving averages (via Lemma 1), the $i$th max-update $||u_n^{(i)}||_\infty$ can be bounded as*

$$||u_n^{(i)}||_\infty \leq \frac{\sqrt{1-\beta_i^{n+1}}}{1-\gamma_i^{n+1}} \frac{1-\gamma_i}{\gamma_i} \sqrt{\frac{\beta_i}{1-\beta_i}} B_n(\beta_i, \gamma_i), \quad i = 1, \ldots, k \qquad (10)$$

*with $B_n$ as defined in Theorem 1.*

We can view this result as placing stronger stability guarantees on $k$-Adam (for $k > 1$) compared to Adam. For example, 2-Adam updates by $u_n^{(2)} = \mathcal{A}_{\gamma_2, \beta_2}(u_{0:n}^{(1)})$, an adaptive normalization of a provably bounded update history $u_{0:n}^{(1)}$ (due to Equation (10)), whereas Adam updates by an adaptive normalization of raw gradients $g_{0:n}$, with $g_{0:n}$ possessing no particular theoretical guarantees. As a result, we would expect $k$-Adam to exhibit more stable updates (though this may not necessarily correspond with improved generalization, which is the purpose of our evaluations below).

**Hyperparameter choice.** For evaluating $k$-Adam, we will consider 4 different strategies for choosing hyperparameters $(\beta_{1:k}, \gamma_{1:k})$,

| | | | |
|---|---|---|---|
| Inverse exp: | $\beta_i := 1 - (1 - \tilde{\beta})^{1/k},$ | $\gamma_i := 1 - (1 - \tilde{\gamma})^{1/k}$ |
| Exp: | $\beta_i := \tilde{\beta}^k,$ | $\gamma_i := \tilde{\gamma}^k$ |
| Scaled: | $\beta_i := \tilde{\beta}/k,$ | $\gamma_i := \tilde{\gamma}/k$ |
| Naive: | $\beta_i := \tilde{\beta},$ | $\gamma_i := \tilde{\gamma}$ |

for all $i = 1, \ldots, k$, where $(\tilde{\beta}, \tilde{\gamma}) := (0.999, 0.9)$ are the typical Adam/AdamW hyperparameter values. In Appendix O we motivate the inverse exp strategy; the other strategies have been chosen heuristically. We note that for these strategies, $(\beta_i, \gamma_i) \in \mathcal{B}_+$ for all $i = 1, \ldots, k$ (since $\beta_i > \gamma_i$) hence we expect stable training as a result of Theorem 3.

**Evaluating $k$-Adam.** We motivated interpreting the implicit effect of scale invariance based on its beneficial influence on training (Tanaka & Kunin, 2021; Lubana et al., 2021; Santurkar et al., 2019). We would therefore expect 2-Adam, and perhaps $k$-Adam for $k > 2$, to outperform Adam; is this indeed the case?

We train a CNN (architecture details in Appendix H) on the CIFAR10 dataset for 100 epochs using $k$-Adam, for values $k = 1, \ldots, 10$ and using each hyperparameter strategy defined above. We display the results in Figure 6, finding that $k$-Adam outperforms Adam/AdamW at various $k > 1$ under the first three strategies, however the naive strategy performs particularly

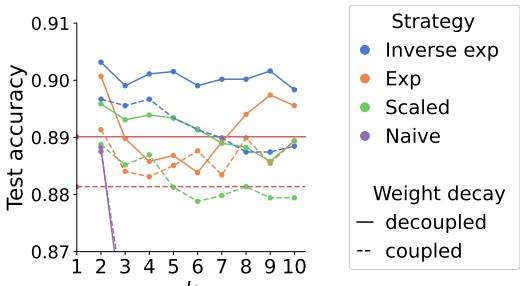

Figure 6: **Optimization with $k$-Adam optimizer.** Plot of the best test accuracy against $k$ after training a CNN for 100 epochs on CIFAR10 using the $k$-Adam optimizer. We highlight the $k = 1$ case in red, corresponding to Adam/AdamW.

badly for $k > 2$; we show this in more detail in Appendix O. We find that 2-Adam with the inverse exp strategy and decoupled weight decay performs best. We also evaluate $k$-Adam on a larger-scale language modeling task in Appendix O, finding similarly that 2-Adam with the inverse exp strategy performs best, outperforming Adam. Though we find $k = 2$ to perform best in this case, it is possible that a larger $k$ may be beneficial in certain contexts, e.g. at small batch sizes (to possibly alleviate noisy gradients). There may also be a more natural, theoretically-motivated hyperparameter strategy compared to the strategies we consider here. We leave a study of these aspects to future work.

## 5 RELATED WORK

**Theoretical analysis of adaptive optimization.** Previous work has studied adaptive optimization theoretically (Chen et al., 2019; Li & Orabona, 2019; Zhou et al., 2024; Barakat & Bianchi, 2020; da Silva & Gazeau, 2019), focusing on asymptotic convergence rates and hence not covering the results presented in this paper. These works perform a discrete-time analysis, except for (da Silva & Gazeau, 2019; Barakat & Bianchi, 2020) which utilize the same first-order differential equation representation for $m(t)$ and $v(t)$ that we describe in Lemma 1 (though they do not explicitly solve

these equations), applied towards understanding the convergence rate of Adam. Additional works that take a continuous-time approach to interpreting adaptive optimization include (Wang et al., 2021), which studies adaptive optimization (with *no momentum*) in the specific case of homogeneous neural networks with a logistic loss function, using a continuous-time expression for $v(t)$ equivalent to that of Lemma 1 to show that adaptive methods based on exponential moving-averages provably converge to the max-margin solution. There is also (Chen et al., 2024b), which studies the Lion optimizer from the perspective of Lyapunov functions and finds Lion to implicitly perform optimization under the bound constraint $||x||_\infty \leq 1/\lambda$. We also highlight the work (Malladi et al., 2024) which provides an SDE formulation of Adam that accounts for gradient stochasticity, which we neglect here.

Various works have derived bounds on Adam's update using a discrete-time approach. (Zou et al., 2021; Zhang et al., 2024) show that, under certain conditions, Adam's update can be bounded by a constant (see Lemma A.2 and Lemma 6.5 respectively), and (Xie & Li, 2024) applies a similar approach to bound Adam's cumulative update (see Lemma 4.2). Such works do not describe the exponential growth outside of the stability region that we derive in this paper, nor do they perform an empirical analysis of this phenomena and its relation to generalization.

**Implicit role of scale invariance.** There has been previous work on interpreting the effect of scale invariance on training dynamics from a theoretical perspective (Tanaka & Kunin, 2021; Kunin et al., 2021; Zhao et al., 2023; Lyu et al., 2023; Lubana et al., 2021). Most relevant to our work is (Tanaka & Kunin, 2021) which, using a Lagrangian-based approach, uncovers an adaptive influence of scale invariance. However this work does not consider optimizers with an adaptive learning rate, such as RMSprop or Adam, which are more relevant to practical deep learning. Our analysis in Section 4 extends this argument to the case of an adaptive learning rate, where we observe a meta-adaptive effect which we convert into an explicit optimizer, 2-Adam.

**Extensions to Adam.** Various extensions to Adam have been proposed. AMSGrad (Reddi et al., 2019) extends Adam by scaling the learning rate by an adaptive factor $1/\sqrt{\hat{v}_n}$ where $\hat{v}_n$ is a maximum of all past moving-averages $v_n$. LAMB (You et al., 2020) makes use of layer-wise adaptive learning rates and a trust ratio mechanism. Lion (Chen et al., 2023) was discovered by an evolutionary algorithm and uses the *sign* of a moving-average of the gradient in order to update the parameters. $k$-Adam extends Adam uniquely, applying an adaptive normalization procedure $k$ times in succession, motivated by a theoretical analysis of how scale invariance influences Adam's training dynamics.

## 6 DISCUSSION

In this paper, we have aimed to highlight the utility of continuous-time frameworks for understanding practical aspects of training dynamics. Our analysis motivated a natural quantity $C(\beta, \gamma)$ relating to training stability which we found to correlate well with generalization – observing that choices of $(\beta, \gamma)$ with larger $C(\beta, \gamma)$ along a given normal curve achieve better generalization performance – providing justification to practical hyperparameter selection. Furthermore, our analysis of training dynamics motivated the 2-Adam optimizer that allows us to exploit the implicit benefits of scale invariance, finding preliminary evidence of 2-Adam outperforming Adam. Our findings are independent of any specific loss function, architecture, or data distribution, allowing for a potentially wide applicability of our approach.

**Limitations and future work.** We have seen that the derived bound on the max-update (Equation (5)) is surprisingly accurate to empirical training dynamics, however our bound does not rigorously justify (only suggests) why we observe a faster rate of generalization for larger $C(\beta, \gamma)$ along a given normal curve. Justifying this observation is an interesting future direction towards a complete understanding of adaptive hyperparameter choice, as well as understanding how stochasticity influences bounding $||u_n||_\infty$ and our findings on the implicit effect of scale invariance. We note that the meta-adaptive effect we describe in this paper is not a complete account of the benefits of normalization layers. Normalization layers possess additional benefits, such as avoiding rank collapse (Daneshmand et al., 2020; Dong et al., 2023; Noci et al., 2022) and contributing towards the smoothness of the loss landscape (Santurkar et al., 2019; Kohler et al., 2018; Karakida et al., 2019; Lyu et al., 2023); it is unclear whether such phenomena are related to the meta-adaptive effect we describe in this paper.

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

## A   DESCRIPTIONS OF OPTIMIZERS

Momentum (Polyak, 1964) features an adaptive gradient direction – updating by a moving-average of the gradient – as described by the update rule,

$$\text{Momentum:}\quad \theta_{n+1} := \theta_n - \eta m_n,$$

$$m_n := \gamma m_{n-1} + (1 - \gamma)g_n$$

for $n = 0, 1, 2, \ldots$, with moving-average hyperparameter $\gamma$ and $m_{-1} := 0$.

RMSprop (Tieleman & Hinton, 2012) features an adaptive learning rate – normalizing the update gradient $g_n$ by a moving-average of the squared gradient $g_n^{\odot 2}$ – as described by the update rule,

$$\text{RMSprop:}\quad \theta_{n+1} := \theta_n - \eta \frac{g_n}{\sqrt{v_n}},$$

$$v_n := \beta v_{n-1} + (1 - \beta)g_n^{\odot 2}$$

with $v_{-1} := 0$. Note that $g_n^{\odot 2}$ describes an element-wise squaring, and the division $g_n/\sqrt{v_n}$ is also element-wise. Here we have neglected weight decay; we will discuss weight decay strategies (coupled vs decoupled) below.

Adam (Kingma & Ba, 2017) and its variant AdamW (Loshchilov & Hutter, 2019) are a combination of momentum and RMSprop, and also use a bias correction factor, which we justify in Appendix B. Neglecting weight decay, Adam & AdamW are equivalent, with update rule

$$\text{Adam/AdamW:}\quad \theta_{n+1} := \theta_n - \eta \frac{\tilde{m}_n}{\sqrt{\tilde{v}_n}},$$

$$\tilde{m}_n := \frac{1}{1 - \gamma^{n+1}}m_n, \quad \tilde{v}_n := \frac{1}{1 - \beta^{n+1}}v_n,$$

$$m_n := \gamma m_{n-1} + (1 - \gamma)g_n, \quad v_n := \beta v_{n-1} + (1 - \beta)g_n^{\odot 2}$$

where a tilde denotes bias correction. The difference between Adam and AdamW comes from how they apply weight decay: Adam uses *coupled* weight decay, equivalent to transforming the loss $L(\theta) \mapsto L(\theta) + \frac{\lambda}{2}||\theta||^2$ such that $g_n \mapsto g_n + \lambda\theta_n$, and AdamW uses *decoupled* weight decay, which involves subtracting $\lambda\eta\theta_n$ from the RHS of the update rule, i.e.

$$\text{AdamW:}\quad \theta_{n+1} := \theta_n - \lambda\eta\theta_n - \eta\frac{\tilde{m}_n}{\sqrt{\tilde{v}_n}}$$

We note that the PyTorch implementation of RMSprop uses coupled weight decay, i.e. RMSprop is Adam with $\gamma = 0$.

## B   MOTIVATING BIAS CORRECTION IN ADAM

Consider an exponential moving average of a sequence $\{x_0, x_1, x_2, \ldots\}$ of tensors,

$$\begin{aligned} y_n &:= \beta y_{n-1} + (1 - \beta)x_n \\ &= \cdots \\ &= (1 - \beta)(x_n + \beta x_{n-1} + \cdots + \beta^n x_0) \end{aligned}$$

Consider the stationary case with $\mathbb{E}[x_n]$ independent of $n$, then

$$\mathbb{E}[y_n] = \mathbb{E}[x_n](1 - \beta)(1 + \beta + \cdots + \beta^n) = \mathbb{E}[x_n](1 - \beta^{n+1})$$

hence if we want an unbiased exponential moving average, we should instead consider a *bias-corrected* form of $y_n$:

$$\tilde{y}_n := \frac{1}{1 - \beta^{n+1}}y_n$$

such that

$$\mathbb{E}[\tilde{y}_n] = \mathbb{E}[x_n]$$

# C  CONTINUOUS-TIME EXPRESSIONS FOR ADAM

*Proof of Lemma 1.*  In the following we will consider the continuous-time limit of small/infinitesimal learning rate $\eta \rightarrow 0$, as done in previous works (Malladi et al., 2024; da Silva & Gazeau, 2019; Barakat & Bianchi, 2020). Given this assumption, we can Taylor expand the definition of $m_n$ in continuous-time as follows,

$$m(t_n) = \gamma m(t_n - \eta^p) + (1 - \gamma)g(t_n)$$
$$= \gamma m(t_n) - \eta^p \gamma \dot{m}(t_n) + (1 - \gamma)g(t_n) + O(\eta^{2p})$$

To order $\eta^p$, $m(t)$ satisfies

$$\dot{m}(t) + \frac{1 - \gamma}{\eta^p \gamma}m(t) = \frac{1 - \gamma}{\eta^p \gamma}g(t)$$

$$\implies \frac{d}{dt}\left[\exp\left(\frac{1 - \gamma}{\eta^p \gamma}t\right)m(t)\right] = \frac{1 - \gamma}{\eta^p \gamma}\exp\left(\frac{1 - \gamma}{\eta^p \gamma}t\right)g(t)$$

$$\implies m(t) = \frac{1 - \gamma}{\eta^p \gamma}\int_0^t d\tau \; \exp\left(-\frac{1 - \gamma}{\eta^p \gamma}(t - \tau)\right)g(\tau)$$

$$=: \int_0^t d\tau \; K_\gamma(\tau, t)g(\tau)$$

Similarly, up to order $\eta^p$, $v(t)$ satisfies

$$\dot{v}(t) + \frac{1 - \beta}{\eta^p \beta}v(t) = \frac{1 - \beta}{\eta^p \beta}g(t)^{\odot 2}$$

$$\implies v(t) = \frac{1 - \beta}{\eta^p \beta}\int_0^t d\tau \; \exp\left(-\frac{1 - \beta}{\eta^p \beta}(t - \tau)\right)g(\tau)^{\odot 2}$$

$$= \int_0^t d\tau \; K_\beta(\tau, t)g(\tau)^{\odot 2}$$

as required.  $\square$

*Proof of Proposition 2.*  We can obtain a differential equation for AdamW by noting that $\theta_{n+1} - \theta_n = -\eta(u_n + \lambda\theta_n)$ and

$$\theta_{n+1} - \theta_n \equiv \theta(t_n + \eta^p) - \theta(t_n) = \eta^p \dot{\theta}(t_n) + \frac{1}{2}\eta^{2p}\ddot{\theta}(t_n) + O(\eta^{3p})$$

$$\implies \eta\lambda\theta(t_n) + \eta^p \dot{\theta}(t_n) + \frac{1}{2}\eta^{2p}\ddot{\theta}(t_n) + O(\eta^{3p}) = -\eta u(t_n)$$

by Taylor expansion, and so in special case of $p = 1$,

$$\lambda\theta(t) + \dot{\theta}(t) + \frac{1}{2}\eta\ddot{\theta}(t) + O(\eta^2) = -u(t)$$

so, up to order $\eta$, we arrive at Proposition 2 as required.  $\square$

**Parameter dynamics when $u(t) \equiv 0$.** We note that in the case of $u(t) \equiv 0 \; \forall \; t$, there are still non-trivial parameter dynamics resulting from Proposition 2 as a consequence of weight decay and a truncated continuous-time expansion. Particularly, in this case, $\theta(t)$ satisfies

$$\lambda\theta(t) + \dot{\theta}(t) + \frac{1}{2}\eta\ddot{\theta}(t) = 0 \implies \theta(t) = ae^{-k_+t} + be^{-k_-t}$$

for $a, b \in \mathbb{R}$ (determined by initial conditions $\theta(t_0), \dot{\theta}(t_0)$) and defining

$$k_\pm := \frac{1}{\eta} \pm \sqrt{\frac{1 - 2\lambda\eta}{\eta^2}}$$

with $k_\pm \geq 0$, assuming that $0 \leq 2\lambda\eta < 1$ (satisfied under typical hyperparameter choice). It is expected that the presence of weight decay would induce non-trivial dynamics even when $u(t) \equiv 0$, since weight decay acts as an additional $\frac{1}{2}\lambda||\theta||^2$ term in the loss function. In the case of no weight decay ($\lambda = 0$), note that $k_- = 0$ with

$$\theta(t) = ae^{-k_+ t} + b$$

This expression isolates the effect of truncating to second-order in Proposition 2. As a result, we have constant dynamics as expected iff $\dot\theta(t_0) = 0$ for some $t_0$ (which implies $a = 0$), with no specific condition on $\theta(t_0) \equiv b$.

**Weight decay.** Note that extending Equation (3) to include weight decay is simple: if we want coupled weight decay (as in Adam), we can transform $g(\tau) \mapsto g(\tau) + k\theta(\tau)$; if we want decoupled weight decay (as in AdamW), we can subtract $k\eta\theta_n$ from the RHS of Equation (1) and proceed identically.

## D   DERIVING THE MAX-UPDATE BOUND

*Proof of Theorem 1.* Using the continuous-time expressions for $m(t)$ and $v(t)$ from Lemma 1, we can write

$$\frac{m(t)}{\sqrt{v(t)}} = \eta^{-p/2}\frac{1-\gamma}{\gamma}\sqrt{\frac{\beta}{1-\beta}}\frac{\int_0^t d\tau\ \exp\left(-\frac{1-\gamma}{\eta^p\gamma}(t-\tau)\right)g(\tau)}{\sqrt{\int_0^t d\tau\ \exp\left(-\frac{1-\beta}{\eta^p\beta}(t-\tau)\right)g(\tau)^{\odot 2}}} \tag{11}$$

with division being elementwise. This expression is particularly amenable to the Cauchy-Schwarz inequality, which says that for arbitrary square-integrable functions $f, h \in L^2(\mathbb{R})$,

$$\left|\int d\tau\ f(\tau)h(\tau)\right| \leq \sqrt{\int d\tau\ f(\tau)^2 \int d\tau\ h(\tau)^2} \tag{12}$$

We wish to derive a bound on Equation (11) that is independent of the specific choice of loss function and architecture, i.e. a bound independent of $g$. We can use the Cauchy-Schwarz inequality to do exactly this, applying $|\cdot|$ elementwise to Equation (11) and bounding its numerator as

$$\left|\int_0^t d\tau\ \exp\left(-\frac{1-\gamma}{\eta^p\gamma}(t-\tau)\right)g(\tau)\right| = \left|\int_0^t d\tau\ \underbrace{\left[\exp\left(-\frac{C(t-\tau)}{2\eta^p}\right)\right]}_{f(\tau)\text{ in Equation (12)}}\underbrace{\left[\exp\left(-\frac{1-\beta}{2\eta^p\beta}(t-\tau)\right)g(\tau)\right]}_{h(\tau)\text{ in Equation (12)}}\right|$$

$$\leq \sqrt{\int_0^t d\tau\ \exp\left(-\frac{C(t-\tau)}{\eta^p}\right)}\sqrt{\int_0^t d\tau\ \exp\left(-\frac{1-\beta}{\eta^p\beta}(t-\tau)\right)g(\tau)^{\odot 2}}$$

$$= \sqrt{\frac{\eta^p}{C}(1-\exp(-Ct/\eta^p))}\underbrace{\sqrt{\int_0^t d\tau\ \exp\left(-\frac{1-\beta}{\eta^p\beta}(t-\tau)\right)g(\tau)^{\odot 2}}}_{\text{equal to denominator of Equation (11)}} \tag{13}$$

defining $C := (2\beta(1-\gamma) - \gamma(1-\beta))/\beta\gamma$. Since this is a bound on the numerator of Equation (11), and the rightmost factor in Equation (13) is exactly equal to the denominator of Equation (11), all dependence on $g$ cancels out when bounding Equation (11) using Equation (13). Applying $||\cdot||_\infty$ to Equation (11) and using Equation (13) results in:

$$\left|\left|\frac{m(t)}{\sqrt{v(t)}}\right|\right|_\infty \leq \frac{1-\gamma}{\gamma}\sqrt{\frac{\beta}{1-\beta}}\sqrt{\frac{1-\exp(-Ct/\eta^p)}{C}}$$

$$\leq \frac{1-\gamma}{\gamma}\sqrt{\frac{\beta}{1-\beta}}\begin{cases}1/\sqrt{C}, & C > 0,\\ \sqrt{t/\eta^p}, & C = 0,\\ \sqrt{\exp(|C|t/\eta^p)/|C|}, & C < 0\end{cases}$$

$$\implies ||u(t)||_\infty \leq \frac{\sqrt{1-\beta^{1+t/\eta^p}}}{1-\gamma^{1+t/\eta^p}}\frac{1-\gamma}{\gamma}\sqrt{\frac{\beta}{1-\beta}}\begin{cases}1/\sqrt{C}, & C > 0,\\ \sqrt{t/\eta^p}, & C = 0,\\ \sqrt{\exp(|C|t/\eta^p)/|C|}, & C < 0\end{cases} \tag{14}$$

Setting $t = t_n \equiv n\eta^p$ and using the definition of $u_n$, we arrive at Theorem 1. As desired, this bound is *independent* of the loss function and architecture. □

**Special cases of RMSprop and signSGD.** When $\gamma = 0$, Adam reduces to RMSprop, and when both $\gamma = \beta = 0$, Adam reduces to signSGD (the update of which is clearly bounded by construction). However, since Theorem 1 requires $\beta, \gamma \in (0, 1)$, our approach in the proof above to bounding Adam's update does not extend to these cases. This is because, to apply Cauchy-Schwarz (Equation (12)), we require square-integrable kernels, i.e. $K_\beta, K_\gamma \in L^2(\mathbb{R})$. However, $K_0(\tau, t) = \delta(t - \tau)$ is not square-integrable ($\int_\mathbb{R} dx\, |\delta(x)|^2 = \infty$), meaning that we require both $\beta, \gamma$ to be non-zero in order for the approach in the above proof to work. In the case of RMSprop, this means that we require a non-zero momentum effect ($\gamma > 0$) in order to provably bound its updates (though, too large $\gamma$ can move dynamics into the unstable region, as demonstrated in Section 3.2).

## E   EMPIRICAL ANALYSIS OF $\mathcal{B}_0$

Equation (5) suggests that $||u_n||_\infty$ may scale like $\sqrt{n}$ when $(\beta, \gamma) \in \mathcal{B}_0$. Do we observe this in practice? In Figure 7 we instead see bounded behaviour analogous to that of the region $\mathcal{B}_+$ (as seen in Figure 3), i.e. there is bounded growth rather than a $\sqrt{n}$ scaling in the max-update. We note that the bound Equation (5) is not violated; the predicted upper bound is indeed met, though the prediction is just not *tight* relative to the true dynamics empirically. We suspect that the condition $C(\beta, \gamma) = 0$ may be sensitive to numerical precision issues in practice, which is perhaps the reason why we observe behaviour analogous to $C(\beta, \gamma) > 0$ instead.

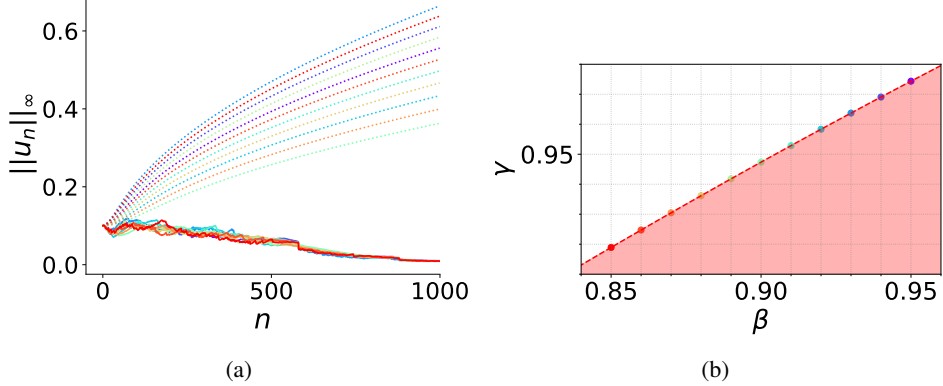

(a)                                          (b)

Figure 7: We plot the max-update over training for 11 points along the boundary $\mathcal{B}_0$. The theoretically predicted $\sqrt{n}$-like scaling is denoted by dotted lines. We use the same experimental setup as used to produce Figure 3, which we describe in Appendix H.

## F   NORMAL CURVES TO $C(\beta, \gamma)$

The level curve $\mathcal{B}_c$ is determined by

$$f(\beta, \gamma) = c, \quad f(\beta, \gamma) := \frac{2\beta(1 - \gamma) - \gamma(1 - \beta)}{\beta\gamma} \equiv C(\beta, \gamma)$$

and normal curves $\gamma = \gamma(\beta)$ satisfy

$$\frac{d\gamma(\beta)}{d\beta} = \frac{\partial f / \partial \gamma}{\partial f / \partial \beta}$$

$$\implies \gamma(\beta) = (k - 2\beta^3)^{1/3}$$

for an integration constant $k$. The normal curve passing through $(\beta_0, \gamma_0)$ has integration constant $k = 2\beta_0^3 + \gamma_0^3$.

# G ASSUMPTIONS OF SECTION 4.2

**Assumption 1.** To verify Assumption 1 we consider the transformer model setup described in Appendix H with qk-norm enabled (Dehghani et al., 2023; Gilmer et al., 2023) such that the loss is scale invariant for each query and key matrix. In Figure 8 we plot the absolute inner product $|\langle W(t_{1000}), g(t_n)\rangle|$ for iterations $n = 1, \ldots, 100$, for each query and key matrix $W$ in the model (a total of 32 matrices).

In accordance with Lemma 2, we observe in Appendix G that the inner product converges to $0$ as $n \to 1000$ when qk-norm is enabled, whereas when disabled as in Appendix G, the inner product remains non-zero and less negligible than when qk-norm is active.

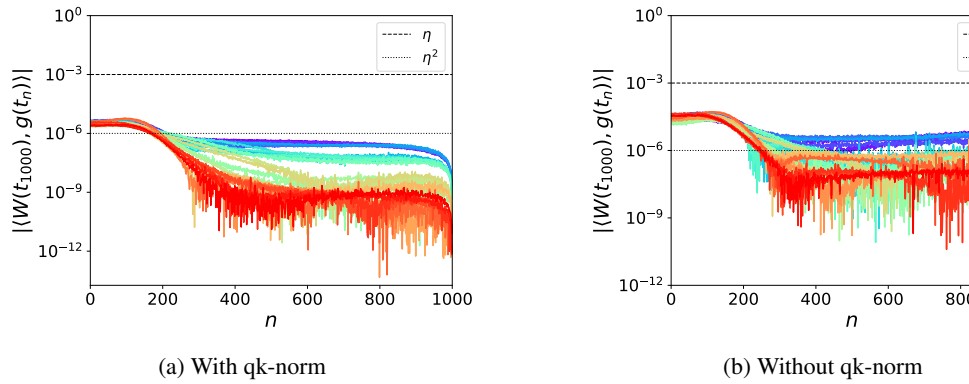

(a) With qk-norm                     (b) Without qk-norm

Figure 8: Plot of the absolute inner product (y-axis) between each query/key matrix and the associated gradient at all previous iterations $n$ (x-axis). In (a) we enable qk-norm, and in (b) we disable qk-norm.

**Assumption 2.** To verify Assumption 2 we choose 16 random parameter values from random query/key matrices $W$, and plot their trajectory in regular training vs. their trajectory when using coarse-graining on $W$. We display the results in Figure 9, finding a strong agreement.

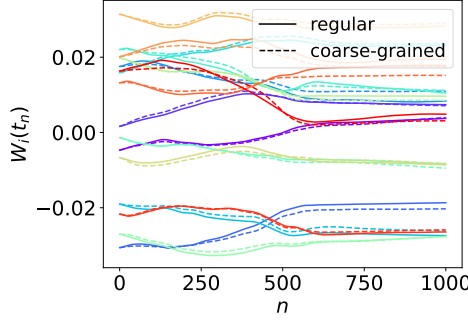

Figure 9: Plot of the trajectories, with and without coarse-graining, for 16 randomly selected parameters from query/key matrices.

# H EXPERIMENTAL SETUPS

**Architecture details.** Throughout the paper we consider a nanoGPT model (Karpathy, 2022) – which is a decoder-only transformer model (Vaswani et al., 2023) – as well as a CNN model (Lecun et al., 1998). The nanoGPT model architecture we use has 6 layers, with 6 attention heads per layer and embedding dimension 384, with a vocab size of 50257, overall possessing 30 million parameters. We use a dropout of 0.2. When training we use the C4 dataset (Raffel et al., 2023), for a max input length of 256 tokens. The CNN model architecture we use is shown in Figure 10 which we use for Figure 6.

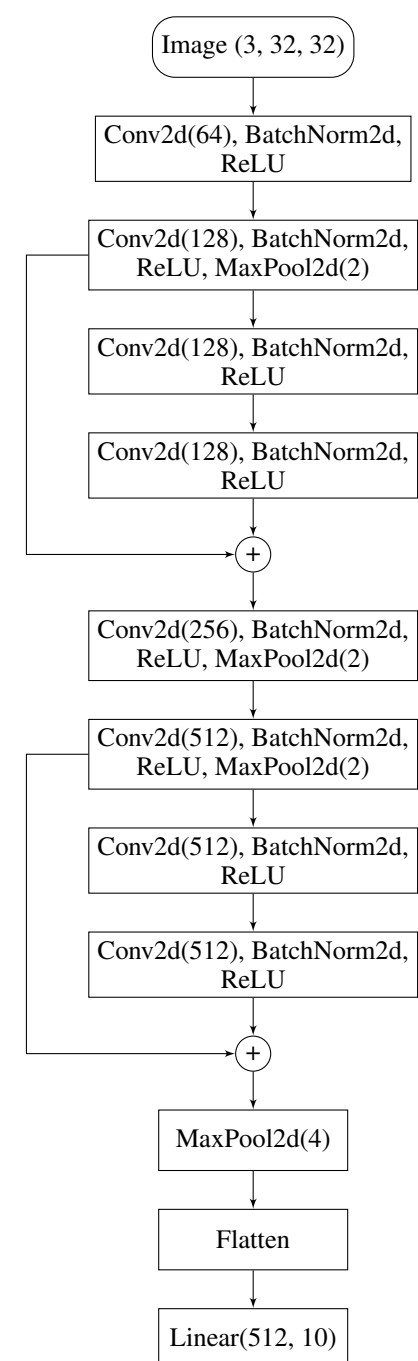

Figure 10: CNN model architecture used in Figure 6. All convolutional layers use a kernel size of $3 \times 3$ and a padding of $1$.

**Continuous-time trajectory.** For experiments that require continuous-time trajectories, we use the method described in Appendix I.

**Figure 1.** We use the method described in Appendix I – with $K = 100$, $p = 1$ and $\eta = 10^{-3}$ – to numerically solve Equation (4), obtaining a continuous-time trajectory in parameter space. We randomly select 16 parameters from a nanoGPT model (trained using setup described above).

**Figure 2.** We describe the equation for normal curves in Appendix C.

**Figure 3.** We train the nanoGPT model using Adam at a learning rate of $10^{-3}$ (and batch size 64) for a range of adaptive hyperparameters $(\beta, \gamma)$ for 3000 iterations, saving the max-update $||u_n||_\infty$ at every iteration $n$. We compute the slope of $\log ||u_n||_\infty$ at $n = 1000$ using the values at $n = 995$ and $n = 1005$.

**Figure 4.** We use the same setup as Figure 3, and additionally use a learning rate warmup (linear, for the first 100 iterations) and after warmup, a cosine decay down to $10^{-4}$. We train for 3000 iterations in order for training to converge such that we can assess generalization performance. We evaluate on a test split of the shakespeare dataset (on 200 iterations worth of test data) every 100 iterations to obtain the test losses.

**Figure 5.** We use the same setup as Figure 1, with $(\beta, \gamma) = (0.999, 0.9)$, training a nanoGPT model for 100 iterations and comparing the predicted value of $||W(t)||^2$ (Equation (8)), computed via the method described in Appendix I, with the true value. We do this for 16 randomly selected query/key matrices.

**Figure 6.** We train a CNN (using architecture in Figure 10) on CIFAR10 for 100 epochs and for each $k \in \{1, \ldots, 10\}$, we sweep over learning rates $\eta \in \{6\text{e-}5, 1\text{e-}4, 3\text{e-}4, 6\text{e-}4, 1\text{e-}3, 3\text{e-}3\}$ and weight decays $\lambda \in \{1\text{e-}6, 1\text{e-}5, 1\text{e-}4, 1\text{e-}3, 1\text{e-}2\}$ and plot the best test accuracy across this sweep. We use a linear warmup for the first 2 epochs, and a cosine decay to $\eta/10$ for the remaining epochs. We evaluate on the test split every 5 epochs. We use $(\tilde{\beta}, \tilde{\gamma}) = (0.999, 0.9)$ and a batch size of 512.

**Figure 16.** We train the nanoGPT model (30M parameters) on the C4 dataset (Raffel et al., 2023) for 20,000 iterations, sweeping over learning rates $\eta \in \{1\text{e-}5, 5\text{e-}5, 1\text{e-}4, 5\text{e-}4, 1\text{e-}3\}$ and weight decays $\lambda \in \{1\text{e-}2, 1\text{e-}3, 1\text{e-}4\}$ to compute the best test loss for each $k \in \{1, \ldots, 5\}$. We restrict to decoupled weight decay to make experiments more computationally tractable.

# I   NUMERICALLY SOLVING EQUATION (4)

**Numerically solving for continuous dynamics.** Defining $\psi(t) := \dot{\theta}(t)$, we can write Equation (4) as two coupled first-order DEs,

$$\dot{\theta}(t) = \psi(t)$$

$$\dot{\psi}(t) = -\frac{2}{\eta^{2p}} \left( \lambda \eta \theta(t) + \eta^p \psi(t) + \eta \frac{\sqrt{1 - \beta^{1+t/\eta^p}}}{1 - \gamma^{1+t/\eta^p}} \frac{m(t)}{\sqrt{v(t)}} \right)$$

and numerically solve via Euler's method,

$$\theta((n+1)\Delta t) \approx \theta(n\Delta t) + \Delta t \psi(n\Delta t)$$

$$\text{with} \quad \psi(n\Delta t) \approx \psi((n-1)\Delta t) + \Delta t \dot{\psi}((n-1)\Delta t)$$

defining $\Delta t := \eta^p / K$. In our numerical validation experiments, we use the timescale $p = 1$ and $K = 100$.

To compute $\dot{\psi}(n\Delta t)$ via the above DE we must compute

$$m(t) = \int_0^t d\tau \, M(\tau, t) g(\tau) = \int_0^{t-\Delta t} d\tau \, M(\tau, t) g(\tau) + \int_{t-\Delta t}^t d\tau \, M(\tau, t) g(\tau)$$

$$= \exp\left( -\frac{1-\gamma}{\eta^p \gamma} \Delta t \right) \left[ \underbrace{\int_0^{t-\Delta t} d\tau \, M(\tau, t - \Delta t) g(\tau)}_{\text{from } n-1\text{th update step}} + \underbrace{\int_{t-\Delta t}^t d\tau \, M(\tau, t - \Delta t) g(\tau)}_{\approx \Delta t \frac{1-\gamma}{\eta^p \gamma} g(t - \Delta t)} \right]$$

## J  IMPLICIT META-ADAPTIVE EFFECT

*Proof of Theorem 2.* In the following we will assume small $\eta$ (as described in Section 2.2), and similarly assume small $\lambda$, treating terms of order $O(\eta\lambda)$ as second-order, which we will neglect (note that the PyTorch default values associated with AdamW are $\eta = 10^{-3}$, $\lambda = 10^{-2}$). Note that we provide empirical validation of this assumption in Figure 5. Applying the inner product $\langle W(t), \cdot \rangle$ to both sides of Equation (7), we arrive at the second-order differential equation

$$\frac{1}{2}\eta\ddot{\varphi} + \dot{\varphi} + 2\lambda\varphi = \eta||\dot{W}||^2 - 2\langle W, u_W \rangle \tag{15}$$

defining the squared norm $\varphi(t) := ||W(t)||^2$. Further, note that $||\dot{W}||^2 = ||u_W||^2 + O(\eta) + O(\lambda)$, since rearranging and squaring both sides of Equation (7) gives

$$||\dot{W}||^2 = ||u_W||^2 + \lambda\eta\langle W, \dddot{W} \rangle + \eta\langle \ddot{W}, u_W \rangle + \frac{\eta^2}{4}||\ddot{W}||^2 + \lambda^2||W||^2 + 2\lambda\langle W, u_W \rangle$$

with $\langle W, u_W \rangle$ second-order in $(\eta, \lambda)$ as a consequence of Assumption 1 and Assumption 2, and further, rearranging and differentiating Equation (15) provides $\ddot{\varphi} = O(\eta) + O(\lambda)$, resulting in

$$\dot{\varphi} + 2\lambda\varphi = \eta||u_W||^2 + O(\eta^2) + O(\eta\lambda) + O(\lambda^2)$$

for a scale invariant weight $W$. Neglecting second-order terms and integrating,

$$||W(t)||^2 = ||W(0)||^2 e^{-2\lambda t} + \eta \int_0^t d\tau \, e^{-2\lambda(t-\tau)} ||u_W(\tau)||^2$$

$$\square$$

*Proof of Proposition 3.* To obtain Equation (9) from Equation (7), we write $W = r\hat{W}$ with $r := ||W||$, which after substituting into Equation (7) becomes

$$(\frac{1}{2}\eta\ddot{r} + \dot{r} + \lambda r)\hat{W} + (\eta\dot{r} + r)\dot{\hat{W}} + \frac{1}{2}\eta r\ddot{\hat{W}} = -u_W \tag{16}$$

Differentiating the condition $\langle \hat{W}, \hat{W} \rangle = 1$ gives $\langle \dot{\hat{W}}, \hat{W} \rangle = 0$, and further, $\langle \ddot{\hat{W}}, \hat{W} \rangle = -||\dot{\hat{W}}||^2$. As a result, applying $\langle \hat{W}, \cdot \rangle$ to Equation (16) gives

$$\frac{1}{2}\eta\ddot{r} + \dot{r} + \lambda r = \frac{1}{2}\eta r||\dot{\hat{W}}||^2 - \langle \hat{W}, u_W \rangle \tag{17}$$

Substituting this expression back into Equation (16) and neglecting second-order terms (noting that, from Equation (17), $\dot{r} = O(\eta) + O(\lambda)$), we arrive at

$$\frac{1}{2}\eta\ddot{\hat{W}} + \dot{\hat{W}} + \Lambda\hat{W} = -\frac{u_W}{r}$$

defining $\Lambda := \frac{1}{2}\eta r||\dot{\hat{W}}||^2$. $\square$

**$||W(t)||^2$ as a moving average.** Equation (8) takes the same form as a moving-average of $||u_W||^2$ in continuous-time. Specifically, consider a generic moving-average of $(x_0, x_1, x_2, \ldots)$,

$$w_n = (1 - \alpha)w_{n-1} + \alpha x_n$$

Then in continuous time, with $w_n = w(t_n)$ and $x_n = x(t_n)$, an identical approach to Appendix C results in the continuous-time expression

$$w(t) = w(0)\exp\left(-\frac{1-\alpha}{\eta^p \alpha}t\right) + \frac{1-\alpha}{\eta^p \alpha}\int_0^t d\tau \, \exp\left(-\frac{1-\alpha}{\eta^p \alpha}(t-\tau)\right)x(\tau)$$

Comparing this expression to Equation (8) allows us to interpret $||W||^2$ as a moving-average of $||u_W||^2$, enabling us to relate $u_W/||W||$ to a form of adaptive normalization in Section 4.2.

## K  CONTINUOUS SYMMETRIES OF THE LOSS FUNCTION

Say $\Phi_s : \Theta \to \Theta$ describes some continuous invariance parameterised by some continuous parameter $s \in \mathbb{R}^S$, i.e.

$$L(\Phi_s(\theta)) = L(\theta) \quad \forall\, s \in \mathbb{R}^S$$

with $\Phi_0(\theta) = \theta \,\forall\, \theta \in \Theta$. Differentiating the above with respect to $s_i$ and evaluating at $s = 0$, we obtain

$$\langle \frac{\partial \Phi_s}{\partial s_i}\Big|_{s=0}, \nabla_\theta L(\theta) \rangle = 0 \quad \forall\, i = 1, \dots, S \tag{18}$$

**Scale invariance.** In the special case of scale invariance with respect to a weight $W$, if we write $\theta = (W, \phi)$ then the relevant transformation is $\Phi_s(\theta) = (sW, \phi)$, and Equation (18) becomes,

$$\langle W, \nabla_W L(\theta) \rangle = 0$$

## L  MAX-UPDATE TRAJECTORY IN $\mathcal{B}_+$

To visualize the theoretical bounds in the case of $(\beta, \gamma) \in \mathcal{B}_+$ in more detail, we plot the results of Figure 3a but restricted to $(\beta, \gamma) \in \mathcal{B}_+$ in Figure 11. We indeed see that the theoretically predicted bounds (dotted lines) are satisfied.

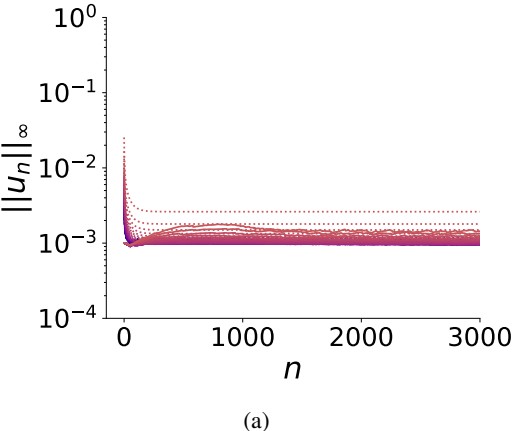

(a)

Figure 11: We plot the results of Figure 3a but restrict to $(\beta, \gamma)$ that lie in $\mathcal{B}_+$.

## M  SUPPORTING DATA FOR SECTION 3.3

In Section 3.2 we found that a larger value of $C(\beta, \gamma)$ along a normal curve correlated with a faster generalization. We further verify this in Figure 12 by instead taking points along the normal curve through $(0.95, 0.9)$, but otherwise an identical experimental setup to the one used to produce Figure 4, finding similar results.

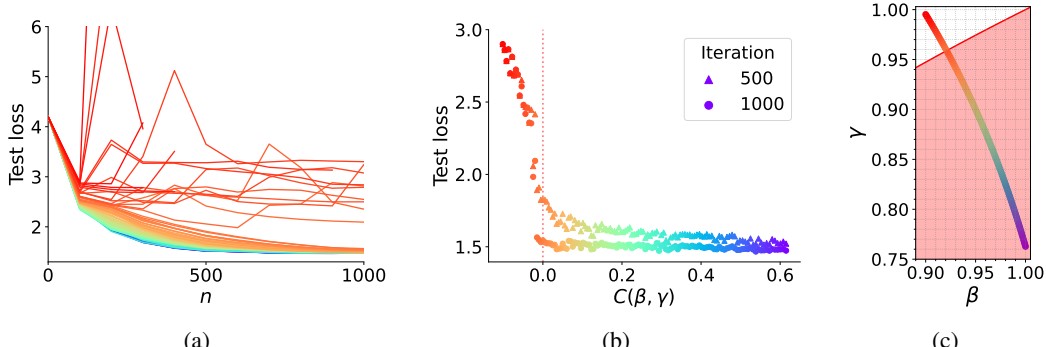

|(a)|(b)|(c)|

Figure 12: Reproduced Figure 4 but instead for the normal curve through the point $(0.95, 0.9)$.

We also consider the level curve $\mathcal{B}_{C(\tilde{\beta},\tilde{\gamma})}$, the results of which we plot in Figure 13. We see a slower rate of generalization for points closer to the origin, as expected, given that such points will have a smaller value of $C(\beta, \gamma)$ relative to other points along their normal curve.

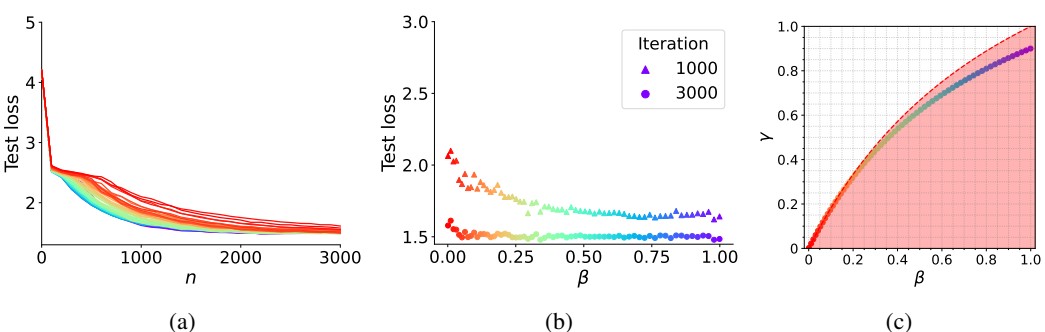

|(a)|(b)|(c)|

Figure 13: Reproduced Figure 4 but instead for the level curve through the point $(0.999, 0.9)$.

## N   ATTEMPTING TO CORRECT EXPONENTIAL GROWTH WITH LEARNING RATE ANNEALING

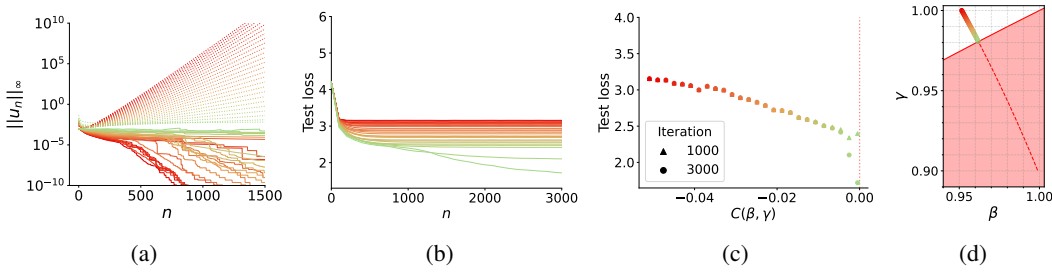

|(a)|(b)|(c)|(d)|

Figure 14: Reproduced Figure 3 and Figure 4 restricted to the region $\mathcal{B}_-$ and using learning rate annealing described below.

Since the parameter update is $\theta_{n+1} - \theta_n \equiv -\eta u_n$, it appears that a well chosen step-dependent learning rate $\eta_n$ may correct the exponential growth in the case of $C(\beta, \gamma) < 0$ described by Equation (5), which occurs via the factor $\exp(n|C(\beta, \gamma)|/2)$. Particularly, given the tightness of the bound seen in Figure 3b, we could consider the learning rate

$$\eta_n = \eta_0 \exp(-n|C(\beta, \gamma)|/2) \tag{19}$$

at step $n$, which results in a bound on $||\theta_{n+1} - \theta_n||_\infty \equiv \eta_n||u_n||_\infty$ that is equivalent to the case of $C(\beta, \gamma) > 0$ with learning rate $\eta_0$. However when we run this learning rate annealing strategy (Equation (19)) in practice, we find that it is too strong, as shown in Figure 14; the max-update becomes very small, and we do not observe good generalization performance. Hence, even though this annealment strategy for $\mathcal{B}_-$ results in the regions $\mathcal{B}_+$ and $\mathcal{B}_-$ possessing the same bounds on $||\theta_{n+1} - \theta_n||_\infty$, their training properties are still quite different. Analysis of weaker annealing methods are out of scope for this work. It is unclear whether correcting such exponential growth in $\mathcal{B}_-$ by choosing an appropriate annealing method would provide any benefits to simply choosing $(\beta, \gamma) \in \mathcal{B}_+$.

## O  CHOICE OF $k$-ADAM HYPERPARAMETERS

The moving-average $m_n^{(k)}$ will have a prefactor of $(1 - \beta_1) \cdots (1 - \beta_k)$, and we can consider a uniform strategy $\beta_1 = \cdots = \beta_k$ and enforce that this prefactor is equal to the typical prefactor for Adam/AdamW, i.e.

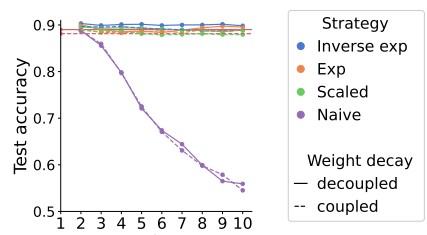

$$(1 - \beta_i)^k = 1 - \tilde{\beta}$$
$$\implies \beta_i = 1 - (1 - \tilde{\beta})^{1/k}$$

**Naive strategy.** In Figure 15 we show a zoomed-out version of Figure 6 to show the performance of the naive strategy across all $k$, emphasizing the importance of well-selected $k$-Adam hyperparameters.

Figure 15: Zoomed out version of Figure 6 to show performance of the naive strategy.

We also found the strategy $\beta_i = \tilde{\beta}^{1/k}$, $\gamma_i = \tilde{\gamma}^{1/k}$ to perform badly.

## P  ADDITIONAL $k$-ADAM EVALUATION

In addition to the image classification experiments of Section 4.3, here we also evaluate $k$-Adam for language modeling on the C4 dataset for a decoder-only transformer of 30 million parameters. We find similar results to Section 4.3: the inverse exp strategy performs best across the strategies, and 2-Adam performs best overall.

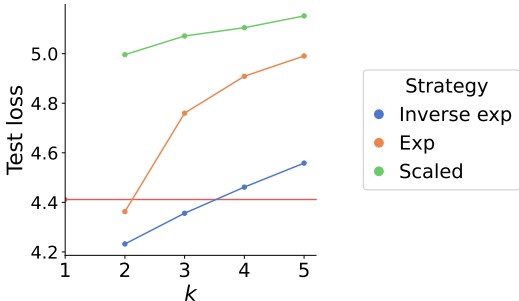

Figure 16: Performance of $k$-Adam for language modelling on the C4 dataset, using a decoder-only transformer with 30 million parameters. We evaluate for $k = 1, \ldots, 5$ and using decoupled weight decay. AdamW's performance is denoted in red. More details regarding experimental setup can be found in Appendix H.