# OpenReview forum: "Continuous-Time Analysis of Adaptive Optimization and Normalization"
_ICLR.cc/2025/Conference — Submitted to ICLR 2025_

### Official Review · Reviewer_R4xU · 2024-10-21

**Soundness:** 2
**Presentation:** 2
**Contribution:** 1
**Rating:** 3
**Confidence:** 5

**Summary:**

In this paper, the authors propose continuous ode approximations for the first and second-order moments $m_t$ and $v_t$. Based on these continuous ode approximations, the authors provide a theoretical guarantee for the stability of the update of Adam. Besides, the authors also demonstrate that the iterates of Adam is an exponential averaging of $m_t/\sqrt{v_t}$ when applied to a scale-invariant architecture. Based on this property, the authors propose a new variant of Adam with a normalization layer.

**Strengths:**

This paper contains some interesting empirical results.

**Weaknesses:**

1. This paper lacks sufficient novelty. Specifically, the conclusion that the update of Adam is bounded given specific exponential moving average rates—one of the major technical contributions of this work—has already been discussed in several prior studies [1, 2, 3, 4], utilizing the Cauchy-Schwarz inequality. Additionally, reference [2] provides a tight bound for the accumulated updates of Adam. Moreover, similar techniques involving continuous approximation for the momentum terms $m_t$ and $v_t$ have also been explored in [5, 6].

2. This paper lacks a thorough discussion of previous works. [6] provides a comprehensive analysis of Lion-K algorithms by constructing an ODE approximation of the dynamic system and establishing a Lyapunov function for this ODE system. By the specific pattern of  the Lyapunov function, they provide an interesting and novel analysis of each hyperparameter in the optimization problem. I believe this contribution is worth highlighting.

3. The assumptions presented in the paper are relatively problematic. Regarding assumption 4.1, while I agree that high-dimensional vectors tend to be nearly orthogonal, the statement that they are "approximately 0" does not imply they are exactly 0. For a rigorous mathematical derivation, the authors should provide a bound for this term and explain why it can be considered negligible in relation to the other terms, rather than simply omitting it as they have done in the current proof. As for assumption 4.2, it is quite unusual. If this assumption is valid, it implies that each coordinate of $v_t$ is equal, which results in the absence of a coordinate-wise adaptive learning rate effect. Under such an assumption, I fail to see any significant distinction between gradient descent with momentum and 'Adam'. Finally, the conclusion that $\\|\dot W_t\\|_2 \approx \\|u_t\\|_2$ based on these two assumptions might also be insuitable. As [2, 3, 7] demonstrated, Adam aligns more with $\ell\_\infty$ norm instead of $\ell_2$ norm.

4. The scaling of Euler's approximation is inconsistent in this paper. In the derivation of ode approximation of $m_t$ and $v_t$, the authors choose the $\eta^p$ as the stepsize and omit all terms containing high order terms, while in the derivation of formula (4), the authors remain a term with coefficient $\eta^{2p}$.

[1] Zou, D., Cao, Y., Li, Y. and Gu, Q. (2023). Understanding the generalization of adam in learning neural networks with proper regularization. In The Eleventh International Conference on Learning Representations, ICLR 2023.

[2] Xie, S. and Li, Z. (2024). Implicit bias of adamw: ℓ∞ norm constrained optimization.

[3] Zhang, C., Zou, D. and Cao, Y. (2024). The Implicit Bias of Adam on Separable Data. arXiv preprint arXiv:2406.10650.

[4] Hong, Y. and Lin, J. (2024). On Convergence of Adam for Stochastic Optimization under Relaxed Assumptions. arXiv preprint arXiv:2402.03982.

[5] Wang, B., Meng, Q., Chen, W. and Liu, T.-Y. (2021). The implicit bias for adaptive optimization algorithms on homogeneous neural networks. In International Conference on Machine Learning. PMLR.

[6] Chen, L., Liu, B., Liang, K. et al. (2023). Lion secretly solves a constrained optimization: As lyapunov predicts. In The Twelfth International Conference on Learning Representations.

[7] Balles, L. and Hennig, P. (2018). Dissecting adam: The sign, magnitude and variance of stochastic gradients. In Proceedings of the 35th International Conference on Machine Learning,

**Questions:**

I suggest that the authors adjust their template since I do not find the indexes of lines in their manuscript.

---

> ### Author Response · Authors · 2024-11-22
> **Rebuttal: response to Reviewer R4xU (1/2)**
>
> We thank the reviewer for their comprehensive feedback on our paper, and for providing us with many helpful references relevant to our work. We appreciate your recognition of our “interesting empirical results”, and in this response we aim to address your central concerns. We have made the following changes towards addressing these concerns:
>
> 1. Provided a more comprehensive review of relevant literature and how our results relate to these works in Section 5, including the references that you have kindly provided.
> 2. Clarified our assumptions more precisely and framed our technical results as lemmas and theorems throughout the paper to aid readability.
> 3. Provided stronger empirical validation of Assumption 4.1 and 4.2 in Appendix G, verifying these assumptions up to 1,000 steps (equivalent to 100,000 continuous-time iterations).
> 4. Refined our presentation of Section 4.2 and 4.3 to make clearer our contributions towards understanding the influence of scale invariance on training dynamics.
>
> We will now address the specific concerns detailed in your review.
>
> > This paper lacks sufficient novelty (...) the conclusion that the update of Adam is bounded given specific exponential moving average rates (...) has already been discussed in several prior studies [1, 2, 3, 4]
>
> Thank you for the references. We agree that the boundedness of Adam’s update under certain conditions has been explored before, however, we believe that the observation of *predictable* exponential growth outside of the bounded region, and an empirical study of the bound’s implications – particularly its tightness to empirical discrete dynamics – is a novel contribution. Furthermore, our subsequent analysis of how scale invariance influences the training dynamics of Adam, as well as the construction of 2-Adam to capture its effect, is a further novel contribution. We also highlight that our analysis is generally applicable, independent of any specific architecture, loss function or data distribution, which contrasts with the analyses in [1, 3]. We have now incorporated these works in Section 5 to provide a more comprehensive literature review and how their contributions relate to our own.
>
> > Moreover, similar techniques involving continuous approximation for the momentum terms $m_t$ and $v_t$ have also been explored in [5, 6]. (...) This paper lacks a thorough discussion of previous works.
>
> Thank you for making us aware of these works. We have now discussed these works in Section 5 of the updated manuscript.
>
> > The assumptions presented in the paper are relatively problematic (...) the statement that they are "approximately 0" does not imply they are exactly 0
>
> We agree that this statement could be more precise, and we have now modified the manuscript to be more explicit in what we mean by this assumption: that the inner product is second-order in $\eta$ and $\lambda$. We have further provided stronger empirical validation of this assumption in Appendix G, validating for up to 1,000 steps (equivalent to simulating 100,000 continuous-time iterations). Further, we note that Figure 5 provides implicit verification of this assumption since the proof of Theorem 2 is dependent upon this assumption. We hope that this point is now clearer, and please let us know if you have any more concerns.
>
> > As for assumption 4.2, (...) I fail to see any significant distinction between gradient descent with momentum and 'Adam'.
>
> We highlight that under assumption 4.2, $\tilde{v}_W(t)$ still depends on time and hence still maintains an *adaptive learning rate* effect that momentum lacks. As a result, we claim that this assumption still captures the important adaptive aspects of Adam relating to a **dynamic learning rate over training**. We also draw your attention to the now stronger empirical verification of this assumption in Appendix G, finding that training dynamics under assumption 4.2 remain close to Adam’s dynamics.
>
> > Finally, the conclusion that $||dot W_t||_2 \approx ||u_t||_2$ based on these two assumptions might also be insuitable. As [2, 3, 7] demonstrated, Adam aligns more with $\ell_{\infty}$ norm instead of $\ell_2$ norm.
>
> Thank you for the references. It appears that these works describe how Adam/AdamW provably converges to the KKT minimizer of an $\ell_{\infty}$ constrained optimization problem, however we fail to see precisely how this finding is related to the norm of the *continuous-time* derivative $||\dot{W}_t||_2 \approx ||u_t||_2$. Could you provide more clarification on exactly how these two concepts are related?

---

> > ### Author Response · Authors · 2024-11-22
> > **Rebuttal: response to Reviewer R4xU (2/2)**
> >
> > > The scaling of Euler's approximation is inconsistent (...) In the derivation of ode approximation of $m_t$ and $v_t$, the authors choose the $\eta^p$ as the stepsize and omit all terms containing high order terms, while in the derivation of formula (4), the authors remain a term with coefficient $\eta^{2p}$.
> >
> > Thank you for spotting this. Throughout the paper we only use equation 4 in the special case of $p=1$, in which case there is a shared factor of $\eta$ that can be divided out of equation 4. This results in us only keeping up to $\eta$ order terms, as seen in equation 7, meaning our approximations are consistent. We have now modified the manuscript to better highlight this.
> >
> > **Summary:** We hope that we have addressed your concerns in our responses and through our additions to the paper. In response to your detailed feedback, we have now strengthened our discussion of related literature and how they relate to our findings, and have further revised Section 4.2 and 4.3 to make our technical results and assumptions clearer. Furthermore, we have strengthened the empirical validation of our central assumptions.
> >
> > We hope that in light of our modifications to the paper, you may be open to revising your score and recommending acceptance of our work. Please let us know if there are any further concerns.

---

> > ### Comment · Reviewer_R4xU · 2024-11-23
> >
> > * The assumptions in Section 4 are both relatively problematical. I'm not satisfied with the authors' response.
> >   * Regarding Assumption 1 in Section 4, as I indicated in my original review, the authors should rigorously derive it is bounded by some term (for example, second-order in $\eta$ and $\lambda$), and rigorously demonstrate why this term can be negligible. The authors do not notice that they already have implicit assumptions: $\eta$ and $\lambda$ are both infinitesimal terms. Based on these implicit assumptions, I do not see any theoretical rationale that directly asserts that all historical terms are of a higher order than these infinitesimal terms without further rigorous illustration. From the perspective of a reader, given the conclusion that $\langle W(t), g_W(t)\rangle = 0$, the assumption $\langle W(t), g_W(\tau)\rangle \approx 0$ implies that $\langle W(t), g\_W(t) - g\_W(\tau) \rangle \approx 0$. Then the natural understanding of the formula above is $g\_W(t) - g\_W(\tau) \approx 0$, which implies $t\approx \tau$, i.e. your training is always at the very initial stage. **I hope the authors can understand that the essence is not how they present such an assumption, the assumption itself is not reasonable from a theoretical perspective.**
> >   * Regarding Assumption 2 in Section 4, I believe I have clearly indicated that **this assumption undermines the coordinate-wise adaptive learning rate effect, which is the most significant distinction between common adaptive methods (including AdaGrad, RMSProp, and Adam) and gradient descent with momentum.** This point has been deliberately omitted by the authors in their response. I do not want to dispute the definition of adaptive methods, as it is not the central issue. However, I firmly believe that under Assumption 2, the algorithm in question cannot be considered Adam; it is merely a normalized version of gradient descent with momentum (GDM). How to handle the momentum term in the denominator is consistently the most critical and challenging aspect in the theoretical analysis of Adam. However, the authors have abandoned the effort to address this technical challenge by proposing a completely impractical Assumption 2. Although I believe that such a strong assumption in theoretical studies should not be based solely on empirical observations, the experimental evidence provided by the authors is also insufficient to support Assumption 2. **Their results can only imply that "Adam" and "Normalized GDM" have similar performance in experiments, instead of each coordinate of $v_t$ are equal. The only empirical evidence is all $v_t$'s indeed have almost equal entries across all coordinates during the whole training process.** (Please do not try to do further experiments to verify whether this is true as I clearly indicated such a strong assumption in theoretical studies should not be based solely on empirical observations.)
> >
> > I understand that the authors may feel frustrated to find that their derived conclusions are overshadowed by earlier related works. However, I still believe it is essential to give appropriate credit to those previous contributions. From my perspective, the current theoretical findings of this paper fall significantly short of the acceptance threshold. I suggest that the authors reconsider the insights of this paper or adjust the focus to an empirical study.
> >
> > [1] Zou, D., Cao, Y., Li, Y. and Gu, Q. (2023). Understanding the generalization of adam in learning neural networks with proper regularization. In The Eleventh International Conference on Learning Representations, ICLR 2023.
> >
> > [2] Xie, S. and Li, Z. (2024). Implicit bias of adamw: ℓ∞ norm constrained optimization.
> >
> > [3] Zhang, C., Zou, D. and Cao, Y. (2024). The Implicit Bias of Adam on Separable Data. arXiv preprint arXiv:2406.10650.
> >
> > [4] Hong, Y. and Lin, J. (2024). On Convergence of Adam for Stochastic Optimization under Relaxed Assumptions. arXiv preprint arXiv:2402.03982.
> >
> > [5] Wang, B., Meng, Q., Chen, W. and Liu, T.-Y. (2021). The implicit bias for adaptive optimization algorithms on homogeneous neural networks. In International Conference on Machine Learning. PMLR.

---

> > > ### Author Response · Authors · 2024-11-29
> > >
> > > We thank the reviewer for pointing us to A.3 of [5] which we have now referenced explicitly in Section 2, and we have also now made an explicit reference to [1, 2, 3, 4] after stating Theorem 1. We reiterate that an empirical study of the bound’s implications, displaying its surprising tightness to discrete updates, is a contribution not present in these referenced works. In regards to Assumption 4.1, we do not follow your reasoning in going from $\langle W(t), g_W(t) - g_W(\tau) \rangle$ being negligible/second-order, to the statement that $g_W(t) \approx g_W(\tau)$. Intuitively, Assumption 4.1 only says that $g_W(\tau)$ lies approximately within the tangent hyperplane to $W(t)$ for all $\tau < t$, which does not necessarily imply that $g_W(t) \approx g_W(\tau)$ (under Assumption 4.1, $g_W(t)$ could be arbitrarily far from $g_W(\tau)$, as long as they both lie within this hyperplane). Could you please further clarify this point?

---

> > > > ### Comment · Reviewer_R4xU · 2024-11-30
> > > >
> > > > Thanks for the follow-up discussions. I would like to first answer your question regarding the Assumption 4.1.
> > > >
> > > > Firstly, I would like to clarify that I do not claim that **$\langle W(t), g_W(t) - g_W(\tau) \rangle\approx 0$ necessarily imply $g_W(t) - g_W(\tau) \approx 0$. However, since you do not provide any theoretical explanation or discussions regarding the rationale of this assumption, the most natural and intuitive understanding of this assumption is $g_W(t) - g_W(\tau) \approx 0$**.  It is straightforward that $g_W(t) - g_W(\tau) \approx 0$ can directly imply that $\langle W(t), g_W(t) - g_W(\tau) \rangle\approx 0$, right? Therefore, from the perspective of the readers, the most direct way to understand why your Assumption 4.1 holds would be $g_W(t) - g_W(\tau) \approx 0$, i.e. $t\approx \tau$. (I notice that another reviewer also raised similar statements.) Actually, my original statement aims to explain why Assumption 4.1 is not convincing to the readers.
> > > >
> > > > Furthermore, I agree that Assumption 4.1 can be guaranteed as long as $g_W(\tau)$ lies approximately within the tangent hyperplane of $W(t)$. However, the question is why you can guarantee that $g_W(\tau)$ lies approximately within the tangent hyperplane of $W(t)$? The authors should notice that compared to the full Euclidean space $\mathbb{R}^{d}$ (here I use $d$ to denote the dimension), the measure of the tangent hyperplane of $W(t)$ is zero. It means that if $g_W(\tau)$ follows any continuous distribution, the probability that $g_W(\tau)$ strictly lies within the tangent hyperplane of $W(t)$ is 0. Based on the discussion above, I also do not feel the assumption that $g_W(\tau)$ lies approximately within the tangent hyperplane of $W(t)$ is reasonable. (Please do not try to dispute whether $g_W(\tau)$ follows continuous distribution or not, this is not the central issue.) I totally understand sometimes we have to assume something in theoretical studies to obtain our conclusions. However, you have to provide solid explanations or evidence to convince your readers why this assumption is acceptable and reasonable.
> > > > For example, if you want to claim that your Assumption 4.1 is mild and acceptable, you can rigorously prove it holds for commonly considered loss functions or data models,  or you can cite other works that also apply such an assumption.
> > > >
> > > > As I stated in my previous response, I totally understand that authors might feel frustrated when to find that their derived conclusions are overshadowed by earlier related works. However, I hope that the authors can also understand that different reviewers might have different evaluation metrics. At least for me, I feel that the theoretical contribution of this paper is relatively insufficient.
> > > >
> > > > Lastly, although I'm confident that the current results of Lemma 1 and Theorem 1 can be directly derived from previous works, and both Assumptions 4.1 and 4.2 are impractical and problematic, I would like to involve in further discussions if the authors could provide more solid theoretical evidence that can verify my points are incorrect.

---

> ### Comment · Reviewer_R4xU · 2024-11-23
>
> I thank the authors for their efforts, while their responses hardly address my concerns.
>
> * I believe this paper has poor theoretical novelty and contribution. After reviewing the revised manuscript and comparing it to the related works I mentioned in my original review, I feel even more confident in this assessment. I list the weaknesses in the following:
>    * As I indicated in my original review, both the conclusion and proof techniques regarding the continuous form of $v_t$ have been proposed in [5], and this can be directly extended to $m_t$ without any modifications. However, in the revised manuscript, the authors still claim that "continuous-time formulation" is their major contribution in their introduction section without even acknowledging [5]. I also did not find any discussion of [5] surrounding Lemma 1 and its proof, which I believe is essential. While I acknowledge the discussion in Section 5, it is not sufficient.  Moreover, if the authors believe the "continuous-time formulation" is their technical contribution, I hope they can provide a detailed illustration of the **essential distinctions between the conclusion and proof technique of Lemma 1 and those presented in section A.3 of [5]** (I know you are using different discretization scale and they consider updating rules with stability constant $\epsilon$, however, neither of these points constitutes an essential difference).
>   * As I mentioned in my original review, the conclusion that the update of Adam is bounded given specific exponential moving average rates, has been widely utilized in theoretical studies regarding Adam [1, 2, 3, 4]. **Specifically, I would like to emphasize that the conclusion regarding the rates approaching infinity outside stable regions can also be directly obtained without additional calculations.** Take the proof of Lemma 6.5 in [3] as an example, they derived that
> $$\frac{m_t[k]}{\sqrt{v_t[k]}} \leq \Big(\sum_{\tau=0}^t \frac{\beta_1^{2\tau}(1-\beta_1)^2}{\beta_2^\tau(1-\beta_2)}\Big)^\frac{1}{2}.$$
> It is straightforward that when $\beta_1^2\leq \beta_2$, the RHS will be bounded by a constant. When $\beta_1^2=\beta_2$, the RHS will be bounded at the scale $\sqrt t$, matching your results $\sqrt n$. And $\beta_1^2>\beta_2$, the RHS will be bounded at the scale of $a^t$ with $a>1$, matching your exponential growth results. Although [1, 2, 3, 4] do not directly present results concerning the unbounded growth rate, as their studies do not focus on this aspect, it does not mean that their findings cannot imply such conclusions. This is evident because they all employ the same proof techniques as you as I mentioned in my original review. Specifically, these works consider practical discretized updates, whereas you are examining an idealized continuous update. This difference suggests that their results are more practical and generalizable. **Moreover, I would like to point out that [1, 3] specify the architecture because their focus differs from that of this paper; however, their derivation for the conclusions mentioned above does not depend on any specific architecture or loss function.**
>   *  I do not think any theoretical conclusion in Section 4 is convincing to me, as they are derived based on two problematical assumptions, which will be discussed in detail.

---

### Official Review · Reviewer_GyMg · 2024-10-23

**Soundness:** 2
**Presentation:** 2
**Contribution:** 2
**Rating:** 5
**Confidence:** 4

**Summary:**

This paper studies Adam through the lens of Ordinary Differential Equations (ODEs). The authors utilize this continuous-time model to reveal aspects of Adam's dynamics in the **full-batch setup, i.e. no stochasticity in the gradients**. In particular, they provide theoretical arguments to identify a stability region for the hyperparameters $\beta_1$ and $\beta_2$, which characterizes the maximum size of the Adam updates. Importantly, they provide some experimental validation of their claims and offer a conjecture that correlates generalization to this region. Finally, they study the beneficial impact of scale-invariant architectural components and leverage their insights to introduce a new optimizer, dubbed k-Adam.

**Strengths:**

- **Originality:** The authors present a novel and quantitatively precise stability region for Adam's betas in the non-stochastic, continuous-time regime, filling a gap in the literature. While several papers discuss Adam’s performance under stochastic gradients, this work focuses on a deterministic model and provides insights under simplifying assumptions.

- **Quality:** The theoretical contributions are sound and grounded in ODE analysis. The derivation of a stability region for Adam is supported by both mathematical rigor and experimental validation. The authors maintain an interesting balance between theory and practical insight, though certain aspects (such as stochastic gradients) are left for future exploration.

- **Clarity:** The paper communicates its main theoretical contribution—the stability region for the betas—relatively clearly, though some sections (like the explanation of k-Adam and the role of normalization) could be refined for better comprehension.

- **Significance:** The work provides potentially valuable insights into the dynamics of Adam in a deterministic setting. If this stability region has practical relevance beyond full-batch training, it could offer a useful guideline for tuning $\beta_1$ and $\beta_2$. Additionally, the introduction of k-Adam as a new optimizer inspired by the theoretical results opens avenues for further exploration.

**Weaknesses:**

Since there is no predefined space to provide a "General Comment", I provide it here.

**Overall Comment:**
The paper addresses interesting and mathematically sound questions, but the manuscript quality is lacking in several areas. It needs substantial improvement, and I recommend **rejecting** the paper, with strong encouragement to resubmit once the following issues have been addressed. In particular, I highlight: i) Missing discussion of literature; ii) Missing discussion on contributions: What is the final product? How can this help practitioners? iii) Better visualizations are needed; iv) Unclear whether the continuous-time model was NECESSARY over the discrete-time setup; Unclear discussion regarding normalization; k-Adam does not seem to be properly justified nor properly tested.

I will now provide a "Detailed Feedback" that covers some weak points and incorporates some questions:

**Detailed Feedback:**

The following points are presented in the same order as they appear in the paper. Unfortunately, due to format alterations in the PDF, I cannot reference line numbers, but I will do my best to highlight clear landmarks:

1. **Literature Review:**
   A review of the literature on continuous-time models for optimizers is crucial, with a particular focus on those that cover Adam. Notably, there is no citation of Malladi et al., who derive an SDE model for Adam, thus addressing the stochastic aspect, which is lacking in this paper.

2. **Limitations Section:**
   A brief section outlining the limitations of the approach would be useful. Additionally, a subsection comparing the method to existing literature could provide more clarity.

3. **Merging Contributions:** In "1. Continuous-time formulation (Section 2)", why not merge these two points into a simpler one? It seems to me that you simply derived an ODE for Adam, which also entails the derivation of Adam's update.

4. **Better explanation and context:** Your main contribution seems to be the stability region for the betas. Parameter sets inside that region imply that the maximum increment of Adam is controlled, while outside of it, an exponential explosion is "predicted". i) While this is interesting, it is unclear if it is practically relevant. Does this give practitioners new insights on selecting the betas? Or are they always in the "safe region"? ii) Is it beneficial to avoid such explosions? Adam is often successfully used even when loss spikes are observed. iii) RMSprop and SignSGD are subcases of Adam: Are they covered by this framework? It seems that RMSprop is always in the stable region, so adding momentum to RMSprop might make it unstable. This is a peculiar observation, as momentum is usually thought of as a stability enhancer.

5. Is it possible that betas in the instability region could benefit from better tuning of the learning rate? I conducted a small experiment on a simple convex quadratic and identified your regions experimentally (via visual inspection, not by plotting your lines). I tested this with learning rates spanning three orders of magnitude and 100 values for each beta: 30,000 small runs. I observed that the stability region expands as the learning rate drops. Could you comment on this? Including a graphical representation for a cheap setup like the one described above would be helpful. If you need the details of my experiment, feel free to ask. Expanding this toy example to the stochastic setting while varying noise levels is also crucial: Could noise shrink the stability region? How does weight decay alter this region?

6. When discussing contribution (f), as well as later in the text, the exact contribution is unclear. Since this is a main point, it must be crystal clear.

7. Regarding contribution (g), it is unclear what advantages k-Adam has. While a high-level description is fine, more details are needed about k-Adam’s advantages, both theoretically and empirically.

8. **Eq. 2:** These formulas for $m$ and $v$ are simply the continuous-time versions of their discrete counterparts, which are already known. Are these surprising in any way? Do they reveal something that the discrete-time version does not?

9. **Eq. 4:** Is your second-order ODE equivalent to the formulation in terms of three first-order ODEs (see Eq. 3.3 of Barakat)? Is there an intrinsic advantage to this point of view?

10. **Eq. 4:** It seems that even if the RHS (e.g., $u(t)$) is set to 0, Adam would still follow a non-trivial second-order ODE that can likely be solved explicitly. This suggests some oscillatory dynamics. Could you elaborate? It seems odd that parameters would evolve even if Adam’s increments are 0.

11. **Figure 1:** I suggest using a color-blind-friendly palette and adding line markers. Rather than comparing the trajectories of some weight, you could calculate the average squared error between the entire trajectory of the real weights and that of the ODE and plot that instead. This would give a more global intuition. I also suggest plotting for longer time and different values of $p$.

12. **Max-update bound:** As highlighted earlier, elaborating more on RMSprop and SignSGD is key.

13. **Eq. 5:** Is this bound independent of the learning rate and $p$? My experiments suggest that the stability region enlarges with a smaller learning rate. Reorganizing this as a well-stated proposition might help.

14. **"We highlight that this bound is only possible because...":** Specify that the reason why you can do this is actually because you do not use stochastic gradients. Also, was it absolutely necessary to use the continuous-time model for this bound, or could it have been derived from the discrete-time version expressions of $m$ and $v$?

15. **Figure 3:** The plots could benefit from reordering. Panel (c) is described before (a) in the caption. Clear legends and a color-blind palette would be helpful.

16. **Section 3.3:** Of course, if you leave the stability region, generalization will likely worsen. However, inside the region, is there a "monotonicity" effect? Is it the case that the further inside the region, the better the generalization? If so, can one always get some guidance in selecting the betas? This is a good point for future research.

17. **Section 4.2:** I’ve reread this multiple times, and it is unclear what the message of this section is. If there is a clear takeaway, I suggest highlighting it. Frame technical results as Lemmas, minimize references to the appendix by grouping them and provide a clear paragraph interpreting the results. Experimental validation is already present, which is good.

18. **k-Adam:** I struggle to see the takeaway from this discussion. First, I suggest removing the 2-Adam discussion currently found before Section 4.3. Otherwise, simply incorporate it into the following section where you properly describe k-Adam. Then, I question the purpose of i) introducing a new optimizer based on loosely justified intuition from Section 4.2, ii) not highlighting clearly its advantages over Adam, and iii) not evaluating it on state-of-the-art experiments. I wonder whether this should be in the main paper as a major contribution.

19.  Finally, this analysis is conducted in a deterministic setting: How can it be generalized to cover stochastic gradients? I expect noise to interact in a non-trivial way with all the moving elements of these analyses.

**Of course**, I reserve the right to alter my score based on the discussion we will have during the rebuttal period.

Malladi et al: On the SDEs and Scaling Rules for Adaptive Gradient Algorithms.

**Questions:**

See Weaknesses.

---

> ### Author Response · Authors · 2024-11-22
> **Rebuttal: response to Reviewer GyMg (1/3)**
>
> We thank the reviewer for their in-depth engagement with our work and for providing us with detailed comments and suggestions. We are happy to hear that you found our work to be “sound and grounded” and to “maintain an interesting balance between theory and practical insight”. To address the concerns detailed in your review, we have made the following substantial changes to the paper:
>
> 1. We have provided a more comprehensive review of related literature in Section 5, including how our results relate to the discrete-time approaches of previous works.
> 2. We have discussed our contributions and their practical implications, the limitations of our framework, and future research directions in Section 6.
> 3. Revised our presentation of adaptive normalization and k-Adam in Section 4.2 and 4.3, now more clearly presenting our technical results as lemmas and theorems and highlighting the implications of our findings.
> 4. Evaluated k-Adam on a larger scale language modeling task, including training a 30M parameter transformer model on the C4 dataset, where we again find **2-Adam to outperform Adam** in Appendix P.
> 5. Provided a discussion of how our approach relates to Adam’s special cases, **RMSprop and signSGD**, in Appendix D.
> 6. Discussed the non-trivial dynamics in the case of $u(t) \equiv 0$ in Appendix C.
>
> Below we provide responses to your specific comments and hope that we have addressed your central concerns with our work.
>
> > A review of the literature on continuous-time models for optimizers is crucial, with a particular focus on those that cover Adam. (...) there is no citation of Malladi et al., who derive an SDE model for Adam (...) Additionally, a subsection comparing the method to existing literature could provide more clarity.
>
> Thank you for your helpful feedback and reference to Malladi et al. To address these concerns, we have now expanded our literature review in Section 5 to include Malladi et al. and its relation to our work, as well as additional works that make use of both discrete-time and continuous-time models in the context of adaptive optimization.
>
> > A brief section outlining the limitations of the approach would be useful.
>
> We have now provided a more complete discussion of our approach’s limitations in Section 6, now emphasizing that our analysis neglects stochastic gradients, and that we lack a complete theoretical understanding (it is only suggested from our theoretical analysis) of why larger values of $C(\beta, \gamma)$ correspond with a faster rate of generalization (along a given normal curve). We hope this has addressed your concern, and please let us know if you have any further concerns.
>
> > In "1. Continuous-time formulation (Section 2)", why not merge these two points into a simpler one?
>
> This is a good point, and we have now combined these two sections into Section 2.2. We have also formatted the associated technical results more clearly as lemmas and propositions, which we hope aids readability.
>
> > While this is interesting, it is unclear if it is practically relevant. Does this give practitioners new insights on selecting the betas? Or are they always in the "safe region"?
>
> Thank you for the question. The motivation behind Section 3 was to justify why the hyperparameter values chosen by practitioners result in successful generalization, where we indeed found that commonly chosen hyperparameters lie within the safe region. Furthermore, in pursuing this, we discovered a natural quantity $C(\beta, \gamma)$ that relates to training stability, finding empirically (in Section 3.3) that this quantity correlates strongly with the rate of generalization in the following sense: along a given normal curve, choices of $(\beta, \gamma)$ with a larger value of $C(\beta, \gamma)$ possess a faster rate of generalization. As a result, our framework has provided novel insight into optimal hyperparameter selection through the measure of stability $C(\beta, \gamma)$, which we believe can indeed aid practitioners in hyperparameter choice.
>
> > Is it beneficial to avoid such explosions? Adam is often successfully used even when loss spikes are observed.
>
> This is an interesting question, and indeed was a motivating factor for our study of generalization performance in Section 3.3. In Section 3.3 (particularly Figure 4) we found that hyperparameters outside of the stable region experienced worsened generalization, *monotonically* in their distance from the stability boundary. There may be some specific contexts for which effective generalization can occur outside of the stability region, perhaps under sufficient learning rate annealing, which is an interesting direction for future research.

---

> > ### Author Response · Authors · 2024-11-22
> > **Rebuttal: response to Reviewer GyMg (2/3)**
> >
> > > RMSprop and SignSGD are subcases of Adam: Are they covered by this framework? It seems that RMSprop is always in the stable region, so adding momentum to RMSprop might make it unstable. (...) elaborating more on RMSprop and SignSGD is key.
> >
> > We agree that discussing these special cases is important. As our expression for $u(t)$ (Theorem 1) depends on the condition $\beta, \gamma \in (0, 1)$, our method of deriving the max-update bound does not extend to the cases of RMSprop and SignSGD since in these cases, at least one of $\beta$ or $\gamma$ is zero. We have now included a discussion of this aspect in Appendix D, noting that our approach only holds for non-zero $\beta, \gamma$ due to the requirement of square-integrability. We hope we have addressed your concern, and please let us know if you have any further questions.
> >
> > > Is it possible that betas in the instability region could benefit from better tuning of the learning rate? (...) Is this bound independent of the learning rate and $p$? My experiments suggest that the stability region enlarges with a smaller learning rate.
> >
> > We thank you for your in-depth engagement with our work and the interesting experimental results. Indeed, our bound for $u_n$ (Theorem 1) does not explicitly depend on the learning rate (nor does it depend on $p$, which we have now commented on at line number 197). However, there is an implicit dependence on the learning rate through the continuous-time assumption of a small learning rate, which we believe may be causing this phenomena. It would be greatly appreciated if you could provide the experimental details behind these results such that we can investigate how the region depends on $\eta$, and how it relates to the prediction of Theorem 1.
> >
> > > Specify that the reason why you can do this is actually because you do not use stochastic gradients (...) this analysis is conducted in a deterministic setting: How can it be generalized to cover stochastic gradients?
> >
> > Thank you for your feedback. We agree that this is an important point to emphasize, and we have now provided comments on this aspect (at line numbers 152, 194, 495, 536). We note that our theoretical predictions agree closely with our experimental results which all use a stochastic, mini-batched version of Adam, and so our results still remain relevant in a stochastic setting. Extending our approach using Adam’s SDE representation, as described in Malladi et al., is an interesting direction for future research.
> >
> > > When discussing contribution (f), as well as later in the text, the exact contribution is unclear. Since this is a main point, it must be crystal clear. (...) it is unclear what the message of this section is. If there is a clear takeaway, I suggest highlighting it. Frame technical results as Lemmas, minimize references to the appendix by grouping them and provide a clear paragraph interpreting the results.
> >
> > We apologize for the unclear presentation of Section 4. We have now made efforts in rewriting Section 4.2 and 4.3 to aid readability and interpretation, now clearly framing technical results as lemmas and theorems, and summarizing the implications of these results. We hope that these changes have addressed this important concern, and please let us know if there are any further concerns.
> >
> > > Regarding contribution (g), it is unclear what advantages k-Adam has. While a high-level description is fine, more details are needed about k-Adam’s advantages, both theoretically and empirically. (...) I question the purpose of i) introducing a new optimizer based on loosely justified intuition from Section 4.2, ii) not highlighting clearly its advantages over Adam, and iii) not evaluating it on state-of-the-art experiments
> >
> > Thank you for the helpful feedback. We agree that our presentation of k-Adam could be improved. Towards addressing this issue, we have made the following changes:
> >
> > 1. Rewritten Section 4.2 and 4.3 to better incorporate 2-Adam and strengthen its motivation.
> > 2. Discussed the difference in theoretical guarantees between k-Adam and Adam in Section 4.3, highlighting the stronger stability properties of k-Adam due to bounding guarantees on each intermediate update (now described by Theorem 3).
> > 3. Provided additional empirical validation of k-Adam in a language modeling setting, training on the C4 dataset with a transformer model of 30M parameters, again finding **2-Adam to outperform Adam**.
> >
> > Overall, we believe that our updated manuscript better explains our purpose in introducing k-Adam as an explicit description of how scale invariance influences training dynamics. We hope that these changes have addressed your concerns. Please let us know if you have any additional concerns.

---

> > > ### Author Response · Authors · 2024-11-22
> > > **Rebuttal: response to Reviewer GyMg (3/3)**
> > >
> > > > Is your second-order ODE equivalent to the formulation in terms of three first-order ODEs (see Eq. 3.3 of Barakat)? Is there an intrinsic advantage to this point of view?
> > >
> > > Thank you for pointing us to this equation. We note that Eq. 3.3 of Barakat is not equivalent to our ODE (Proposition 2) as Barakat neglects second-order derivatives in the parameter $\theta$, whereas we maintain these terms. In the limit of infinitesimal learning rate ($\eta \to 0$), one may neglect these second-order derivative terms as done in Barakat, however practical (non-infinitesimal) learning rates (such as the common choice $\eta \approx 10^{-3}$) will introduce a deviation from the continuous model. For our study to be as faithful as possible to practice while maintaining a tractable theoretical analysis, we keep up to second-order derivative terms, and indeed we find this to be sufficient to closely match empirics as seen in Figure 1 and 5. We also note that our analysis of Section 4 includes (decoupled) weight decay, which is an aspect that Barakat neglects. Other than these two aspects, Barakat’s formulation is equivalent to ours. We hope this answers your question, and please let us know if you have any further questions regarding this point.
> > >
> > > > It seems that even if the RHS (e.g., $u(t)$) is set to 0, Adam would still follow a non-trivial second-order ODE that can likely be solved explicitly. This suggests some oscillatory dynamics. Could you elaborate?
> > >
> > > Thank you for the interesting observation. There are two factors involved in these non-trivial dynamics: (1) the presence of weight decay and (2) the use of a truncated (up to second-order derivatives) continuous-time expansion. In Appendix C we have provided a discussion of this aspect, explicitly solving the associated second-order ODE and finding that weight decay and a continuous-time truncation result in non-constant solutions unless specific initial conditions are met. Thank you for the intriguing question, and we hope that our additions have shed light on this aspect.
> > >
> > > > Of course, if you leave the stability region, generalization will likely worsen. However, inside the region, is there a "monotonicity" effect? Is it the case that the further inside the region, the better the generalization? If so, can one always get some guidance in selecting the betas?
> > >
> > > This effect is captured in Figure 4(b), where we find that points $(\beta, \gamma)$ further inside the stability region experience a faster rate of generalization that appears monotonic in the distance of $(\beta, \gamma)$ from the stability boundary. We also provide further evidence of this phenomena along a different normal curve in Appendix M. These findings show that, along a given normal curve, choices of $(\beta, \gamma)$ with a larger value of $C(\beta, \gamma)$ experience a faster rate of generalization, providing a novel theoretically-guided principle towards optimal hyperparameter selection.
> > >
> > > **Summary:** We greatly appreciate the reviewers' extensive engagement with our work and for providing insightful questions and suggestions. Towards addressing the reviewer’s concerns, we have made substantial additions to our paper, including a more comprehensive literature review of continuous-time approaches to optimization, improved empirical validation of k-Adam in a language modeling setting for a 30M parameter transformer model, and a revision of Section 4.2 and 4.3 to better illustrate our contributions and the purpose of k-Adam.
> > >
> > > Given these revisions, we hope that you may consider revising your score. We remain available to address any further questions or concerns.

---

> > > > ### Comment · Reviewer_GyMg · 2024-11-22
> > > > **Thanks**
> > > >
> > > > Dear Authors,
> > > >
> > > > Thank you for taking my review seriously.
> > > >
> > > > On one hand, my original recommendation stated that your paper needed much rework and improvement: I acknowledge that you did improve the paper. On the other, I also have to highlight that such changes are quite significant: If I simply focus on the lines you highlighted in red, you essentially changed a fifth of the paper, which indicates that the paper was not indeed in the shape to be published.
> > > >
> > > > I appreciate the replies to my points, and I only have a few more ones for you to reply to:
> > > >
> > > > 1. Regarding my original point 10, is it possible that your ODE is simply ill-defined because you did not specify the initial conditions? I am very worried by the fact that setting $u=0$ still implies a movement of the parameters.
> > > > 2. A colorblind palette is highly encouraged: I am sorry if I insist on this point, but some people struggle to tell colors and I am likely not the only one --- This impairs the possibility to clearly read your figures and interpret them correctly.
> > > > 3. It is still unclear to me why you could have not derived this bound from the closed-form formulas of $m$ and $v$ that one can easily derive in discrete time.
> > > > 4. The caption of Figure 3 is still in the wrong order w.r.t. the orders of the presented panels.
> > > > 5. I am pretty skeptical about k-Adam: When putting forward an optimizer, one has to perform a large number of experiments, which Is not the case here. Importantly, Adam-k is more expensive than Adam. Therefore, Adam should be allowed for more iterations to make up for its lower computational requirements. Did you make sure to run experiments to ensure this fairness?
> > > >
> > > > To conclude, I still believe that this paper needs to be improved, and it is not ready for publication: **I increase my score to 5**, and will engage with you, the other Reviewers, and the AC in the appropriate moments to help distill the final decision.

---

> > > > > ### Author Response · Authors · 2024-11-23
> > > > > **Thank you for your thoughtful engagement!**
> > > > >
> > > > > To GyMg:
> > > > >
> > > > > Thank you for increasing your score! We further address your concerns below.
> > > > >
> > > > > > Regarding my original point 10, is it possible that your ODE is simply ill-defined because you did not specify the initial conditions? I am very worried by the fact that setting $u=0$ still implies a movement of the parameters.
> > > > >
> > > > > In Appendix C, we demonstrate that when weight decay is absent and given the initial condition $\dot{\theta}(t_0) = 0$ at some time $t_0$, the parameters *do not change* when $u=0$, regardless of the specific value of $\theta(t_0)$. This confirms that the parameter dynamics are well-defined and behave as expected under the initial conditions $\dot{\theta}(0) = 0$ and $\theta(0)$ determined by the initialization scheme. When weight decay is non-zero, the observed parameter movement is natural and expected, since weight decay contributes independently to the loss function, separate from the term involving $u(t)$.
> > > > >
> > > > > > A colorblind palette is highly encouraged: I am sorry if I insist on this point, but some people struggle to tell colors and I am likely not the only one --- This impairs the possibility to clearly read your figures and interpret them correctly.
> > > > >
> > > > > Thank you for raising this important accessibility concern. We have revised all figures in the main text to use a colorblind-friendly palette to ensure they are accessible to all readers.
> > > > >
> > > > > > It is still unclear to me why you could have not derived this bound from the closed-form formulas of $m$ and $v$ that one can easily derive in discrete time.
> > > > >
> > > > > While the bounds on Adam's update can indeed be derived through discrete-time analysis (as discussed in Section 5), our continuous-time approach offers two key advantages. First, it makes the exponential growth phenomena more readily apparent. Second, it provides a natural foundation for our analysis in Section 4, which relies heavily on continuous-time expressions for $m$ and $v$. Although the choice between discrete and continuous-time formulations may seem primarily aesthetic, the continuous framework proved particularly valuable in uncovering the meta-adaptive property of scale invariance, an insight that would be less apparent in a discrete-time approach.
> > > > >
> > > > > > The caption of Figure 3 is still in the wrong order w.r.t. the orders of the presented panels.
> > > > >
> > > > > Thank you for noting this inconsistency. Rather than reordering the panels, we have modified the caption to match the existing panel order to maintain consistency with our other figure references throughout the reviewer responses.
> > > > >
> > > > > > I am pretty skeptical about k-Adam: When putting forward an optimizer, one has to perform a large number of experiments, which Is not the case here.
> > > > >
> > > > > We agree that a comprehensive empirical evaluation would be necessary to propose k-Adam as a practical successor to Adam. However, our introduction of k-Adam serves a different purpose: it provides an explicit, more interpretable description of how scale invariance influences training dynamics, aligning with our paper's theoretical focus. We’d like to highlight that our updated experiments, which use the C4 dataset with a 30M-parameter transformer model, already provide some empirical validation, with Appendix P showing that 2-Adam robustly outperforms Adam in this language modeling setting.
> > > > >
> > > > > Thank you for your thorough and constructive feedback, which has helped us substantially improve our manuscript. We hope that with these changes, you could consider recommending our paper for acceptance. Please don't hesitate to raise any additional questions to ensure complete clarity!

---

> > > > > > ### Comment · Reviewer_GyMg · 2024-11-26
> > > > > >
> > > > > > Dear Authors,
> > > > > >
> > > > > > I appreciate all your efforts, but I am unwilling to go above 5 for the above reasons.
> > > > > >
> > > > > > I am sure that you will improve this paper and get it in a better shape for a submission to another venue.
> > > > > >
> > > > > > Best regards.

---

> > > > > > > ### Author Response · Authors · 2024-11-29
> > > > > > >
> > > > > > > We again thank the reviewer for an in-depth engagement with our work and for providing very helpful feedback!

---

### Official Review · Reviewer_13ZQ · 2024-10-25

**Soundness:** 3
**Presentation:** 3
**Contribution:** 2
**Rating:** 6
**Confidence:** 3

**Summary:**

The author presents a continuous time analysis of Adam. They first present the continuous version of Adam and a corresponding differential equation (eq 4). With this formulation they present a bound on the parameter updates (eq 5) which depends on the adam hyperparamters. They find that this bound holds reasonably well in practice, which provides some justification for why certain Adam hyperparameters work well in practice. The authors also motivate why Adam+normalization behaves like adam applied multiple times, a method they call Adam-k. The authors provide some small-scale experiments with Adam-k, showing benefits for training on CIFAR10.

**Strengths:**

- The paper is well written and crisp.
- Continuous time analysis of Adam is interesting and novel (AFAIK, but I’m not an expert). Since Adam is an important algorithm, any analysis of it can be impactful.

**Weaknesses:**

- Assumptions 1 and 2 are not very intuitive, at least not to me.
- CIFAR10 results are not very convincing, it’s small scale.

**Questions:**

- How tight is the bound of eq 5? Is there a corresponding lower bound?
- Can you add more experimental verification of Adam-k?

---

> ### Author Response · Authors · 2024-11-22
> **Rebuttal: response to Reviewer 13ZQ**
>
> We thank the reviewer for their helpful feedback and positive assessment of our work. We are very happy to hear that our findings were “well written and crisp”, and that our “continuous time analysis of Adam is interesting and novel”. To address your concerns, we have made the following additions to the paper:
>
> 1. **Improved experimental verification of k-Adam**, now evaluating k-Adam in a language modeling setting, training a transformer model with 30M parameters on the C4 dataset and similarly finding **2-Adam to outperform Adam**, with results in Appendix P.
> 2. Provided a more detailed discussion of Assumptions 1 and 2 in Section 4.2, as well as including a stronger empirical validation of these assumptions in Appendix G.
>
> We have also addressed your specific feedback below, which we hope fully addresses your concerns.
>
> > Assumptions 1 and 2 are not very intuitive, at least not to me.
>
> Thank you for your feedback. To aid intuition regarding these assumptions, we have now extended our discussion of both assumptions in Section 4.2. Specifically, we more clearly highlight the orthogonality property of scale invariance (described by Lemma 2) and how it supports Assumption 1, and the interpretation that Assumption 2 corresponds to assuming that every entry in the weight $W$ experiences the same amount of adaptive scaling. We also draw your attention to the stronger empirical verification that we have now provided for these assumptions, which can be found in Appendix G. We hope this addresses your concern. Please let us know if you have any questions regarding these assumptions.
>
> > How tight is the bound of eq 5?
>
> In Figure 3(b) we showcase that, in the case of $C(\beta, \gamma) < 0$, the bound is extremely tight to the true empirical dynamics, with the empirical rate of exponential growth in close agreement with the theoretically predicted rate of $|C(\beta, \gamma)|/2$ (denoted by a red dashed line). In the case of $C(\beta, \gamma) > 0$, we look more closely at the bound in Appendix L, finding it to remain satisfied as expected, and to be reasonably tight.
>
> > Is there a corresponding lower bound?
>
> Thank you for the interesting question. In our analysis we did not determine a lower bound on the max-update $||u_n||_{\infty}$, though we do find empirically that dynamics stay quite tight to the upper bound (as displayed in Figure 3(b) and Appendix L). Exploring this aspect further from a theoretical perspective is an interesting future research direction.
>
> > CIFAR10 results are not very convincing, it’s small scale (...) Can you add more experimental verification of Adam-k?
>
> Thank you for the feedback. We agree that the experimental verification of k-Adam could be stronger. To address this point, we have now evaluated k-Adam in a language modeling setting on the C4 dataset, considering a transformer model of 30M parameters. We include the results in Appendix P, where we again find 2-Adam to outperform Adam. We hope this addition has addressed this concern, and please let us know if you have any further concerns.
>
> **Summary:** We would like to thank the reviewer for providing very helpful suggestions and comments, and for their positive assessment of our findings. On the basis of your feedback, we have provided a stronger assessment of k-Adam’s performance in a larger scale language modeling setting, and have better clarified and verified the assumptions that facilitate our theoretical analysis. We are readily available to address any further concerns.

---

> ### Author Response · Authors · 2024-11-29
> **Rebuttal: reminder to Reviewer 13ZQ**
>
> We would like to gently remind the reviewer that the discussion period ends within the next four days and that we have responded to their review. We hope you will let us know if we have addressed your concerns, and if you have any further concerns, we would be happy to address them.

---

> > ### Comment · Reviewer_13ZQ · 2024-12-01
> >
> > Thanks for your response, I'll retain my score!

---

### Official Review · Reviewer_bK6b · 2024-11-05

**Soundness:** 2
**Presentation:** 2
**Contribution:** 2
**Rating:** 3
**Confidence:** 3

**Summary:**

This paper presents a continuous-time formulation of Adam(W). The authors derive the continuous-time ODE to approximate the parameter dynamics of Adam(W) via a Taylor expansion. The main application of the theory is to predict the stable region of hyperparameters $(\beta, \gamma)$ in which the Adam update won't blow up. The authors also use a similar method to study the dynamics of scale-invariant networks and identify the so-called "meta-adaptive effect," which needs further clarification.

However, I am a bit suspicious regarding the theoretical results of the paper. Most of the derivations are non-rigorous and hand-wavy.  Furthermore, the experimental results lack diversity and scale.

Overall, I think the research direction is interesting and the results can be quite useful. However, the theoretical derivations lack rigor and clarity. I am afraid I don't feel comfortable to support acceptance of the paper in its current form.


------- update -----------
thanks for addressing some of the concerns and updating the paper + experiments;
i raised my score to 5; but i echoed Reviewer GyMg's point, the paper would benefit from further refinement and resubmission.

**Strengths:**

* The paper proposes a framework to approximate the parameter dynamics of Adam(W), which could be potentially very useful.
* This framework can help estimate the stable region of hyperparameters $(\beta, \gamma)$, which can be helpful in practice for hyperparameter selection.
* The framework is relatively easy to understand.

**Weaknesses:**

- The whole theoretical framework is non-rigorous and based on flawed derivations/assumptions.
    - I don't understand the crucial deviation in Section C, and it should be put in the main text.
    - First, why is $m$ differentiable, why can you drop the higher order (second derivative is bounded?) of it?
   -  Second, why is it $g(t_n)$ not $g(t_n - \eta^p)$? In the latter setting, you would need to have an extra term $g'(t_n)$ and an error term $g''(t_n)$?
   - I think these are crucial questions, on which the whole paper is based. I also don't know why we should expect a continuous formation of the dynamics of $m$ in the first place.

* The experimental results are not convincing enough. I would expect larger scale + more diversity experiments, given the theoretical results are not rigorous.
   - The training steps ($n=100$ or $1500$) are too few. should consider $n= O(10k)$
   - The scale and diversity of the experiments are limited (a small transformer on Shakespeare). I would expect the authors to run experiments on a more realistic dataset, like a 20M-100M parameter model on a subset of C4 or similar.

**Questions:**

- It is unclear how to get equation 7.

-  can you clarify what does META-ADAPTIVE EFFECT mean?

- can you clarify why the exponential moving average ($\|W\|$ vs $\|u_W\|$) is significant and surprise ?

- I don't understand this "η  ̈φ = O(η2) + O(λη)," , and why you can drop this term, which hasn't mentioned in the main text.

---

> ### Author Response · Authors · 2024-11-22
> **Rebuttal: response to Reviewer bK6b (1/3)**
>
> Thank you for thoroughly reading our paper and providing us with constructive technical feedback! We are delighted that you found our research on establishing the continuous-time framework of Adam(W) to be "potentially very useful" and our findings on stable hyperparameter regions to be "helpful in practice." We also greatly appreciate your recommendations to clarify and verify the crucial assumptions underlying our theory and your thoughtful suggestions for experiments to fully validate our framework's predictions.
>
> To address your concerns, we have taken the following steps:
>
> 1. We have made the set of theoretical assumptions more explicit and expanded the technical results with greater detail in the main text. We have also updated our draft after incorporating your suggestions.
> 2. We have conducted the experiments you have recommended, including training a transformer model with 30M parameters on the C4 dataset, to validate these assumptions comprehensively.
>
> Below, we detail how we have addressed your feedback, and we hope this fully addresses your concerns.
>
> > I don't understand the crucial deviation in Section C (...) First, why is $m$ differentiable, why can you drop the higher order (second derivative is bounded?) of it?
>
> Thank you for carefully reviewing our derivation in Appendix C. We sincerely appreciate your insightful feedback. First, we would like to clarify that the goal of this section is to approximate the trajectories of the discrete-time update rule using a continuous-time differential equation. Accordingly, the differentiability of $m$ is an assumption we make, rather than a result derived from our analysis, an important point that we should have emphasized and discussed more thoroughly in the main text. Regarding the dropping of the higher order terms, we now explicitly address that we assume $p \geq 1$ throughout the paper and clarify that we consider $\eta$ to be small, consistent with practical implementations (see comment at line number 125). These are indeed crucial assumptions that we agree should have been more explicitly stated.
>
> In response to your feedback, we have made the following updates:
> 1. We now motivate the continuous-time formulation by referencing prior influential works in Section 5 that have derived practically insightful claims under similar assumptions.
> 2. We provide a comment in Section 2.2 (line number 125) explaining why dropping the second-order term is reasonable, particularly given the small learning rate $\eta$ typically used in practical implementations of Adam (e.g., $\eta = 10^{-3}$ by default in its PyTorch implementation, making higher-order terms practically negligible).
> 3. We have substantially increased the step counts in Figures 1 and 5, extending the validation of our theory to 1,000 steps (corresponding to 100,000 continuous-time iterations)—significantly more than the 100 steps in the previous version—to address your suggestions more comprehensively.
>
> Overall, we believe these updates better motivate, clarify, and validate the assumptions in our work, aligning more closely with your suggestions.
>
> > Second, why is it $g(t_n)$ not $g(t_n-\eta^p)$?
>
> This is an interesting point. We take this formulation to align with the standard implementation of Adam (cf. PyTorch).  If we instead defined $m_n = \gamma m_{n-1} + (1-\gamma) g_{n-1}$, such that the continuous-time expression includes $g(t_n-\eta^p)$, we note that updates $u_n$ based on $m_n$ would use old gradients $g_{n-1}$ from one-step behind, with the most recent gradient $g_n$ being neglected. It seems it would be best to update using the most up-to-date parameter value $\theta_n$, and indeed this is what the official PyTorch implementation of Adam does (https://pytorch.org/docs/stable/generated/torch.optim.Adam.html), updating via the gradient associated with the most recent parameter value. We therefore define $m_n = \gamma m_{n-1} + (1-\gamma) g_n$ to accurately match practice, which results in a $g(t_n)$ term rather than $g(t_n-\eta^p)$.

---

> > ### Author Response · Authors · 2024-11-22
> > **Rebuttal: response to Reviewer bK6b (2/3)**
> >
> > > I also don't know why we should expect a continuous formation of the dynamics of $m$ in the first place.
> >
> > We appreciate your concern regarding the expectation of a continuous-time formulation of the dynamics of $m$. Our motivation for pursuing a continuous-time formulation stems from the observation that modern implementations of Adam(W) utilize small learning rates (e.g., $\eta \approx 10^{-3}$). This implies that the leading-order terms in the continuous-time approximation are sufficient to model the behavior observed with discrete-time updates.
> >
> > To verify this intuition, we have conducted empirical validations presented in Figure 1 and Figure 5, which demonstrate that the continuous-time formulation of $m$ accurately models its discrete-time behavior. Additionally, the close agreement between the predictions of Equation 5 and the empirical results shown in Figure 3 provides indirect evidence supporting our claim, as Equation 5 relies on this continuous-time assumption.
> >
> > > The experimental results are not convincing enough. (...) The training steps ($n=100$ or $1500$) are too few (...)  I would expect the authors to run experiments on a more realistic dataset, like a 20M-100M parameter model on a subset of C4 or similar.
> >
> > Thank you for your valuable suggestion. In response, we have updated our draft to fully incorporate your feedback:
> > * **Model Scaling**: We have increased the size of the transformer model from 10 million parameters to 30 million parameters and trained it on the C4 dataset, as you recommended, now for 3,000 iterations rather than only 1,500.
> > * **Extended Simulations**: We have extended the simulation of the continuous-time trajectory to 1,000 steps (equivalent to 100,000 continuous-time iterations)—ten times more than the previous 100 steps—in Figures 1 and 5. This increased step count maintains a close agreement with the discrete dynamics.
> >
> > We have updated the manuscript to reflect these new results. We hope these revisions address your concerns regarding the empirical verification of our assumptions and theoretical derivations.
> >
> > > It is unclear how to get equation 7.
> >
> > We apologize for the confusion. Equation 7 is a direct consequence of equation 4 when $p=1$, which we have now emphasized in the updated manuscript. We have also framed technical results into lemmas and propositions which we hope aids readability.
> >
> > > can you clarify what does META-ADAPTIVE EFFECT mean?
> >
> > Thank you for this clarification. We have now revised Section 4.2 and 4.3 and introduced new notation in Section 2.1 to make this aspect clearer. In summary, we can think of Adam as updating by an *adaptive normalization* $u_n$ of the gradient history $(g_0, …, g_n)$, defining such an *adaptive normalization* as the procedure of taking a moving-average over $(g_0, …, g_n)$, and dividing by the square-root of the moving-average over $(g_0^2, …, g_n^2)$, where squaring is elementwise, resulting in the update $u_n$.
> >
> > By *meta*-adaptive normalization, we are referring to the concept of applying a *further* adaptive normalization procedure, now over the history $(u_0, …, u_n)$ of updates. This involves taking the moving-average over $(u_0, …, u_n)$ and dividing by the square-root of the moving-average over $(u_0^2, …, u_n^2)$, resulting in an update we denote $u_n^{(2)}$. Since this corresponds to applying an adaptive-normalization procedure twice in succession, we refer to it as *meta-adaptive*.
> >
> > This concept is motivated by studying how scale invariance influences Adam’s dynamics, where we find the dynamics of a scale invariant weight corresponds with a meta-adaptive update as described above. This motivates us to define the 2-Adam optimizer, which is an explicit formulation of this concept; an optimizer that updates precisely by $u_n^{(2)}$. k-Adam further generalizes this concept, applying an adaptive-normalization procedure $k$ times in succession. We hope this description, as well as our revisions to the paper, clarify this aspect, and please let us know if you have further questions.

---

> > > ### Author Response · Authors · 2024-11-22
> > > **Rebuttal: response to Reviewer bK6b (3/3)**
> > >
> > > > can you clarify why the exponential moving average ($|W|$ vs $|u_W|$) is significant and surprise ?
> > >
> > > Naively we would think that the presence scale invariance has no relation to adaptive optimization, however, we find that the presence of scale invariance has an *implicit* adaptive optimization effect that acts in combination with the *explicit* adaptive optimization performed by Adam. Specifically, the observation that $|W|^2$ is a moving-average of $|u_W|^2$ (now described by Theorem 2) is significant as it allows us to relate scale invariance to adaptive optimization in this manner.
> > >
> > > In more detail, as seen by Proposition 3, we find that $\hat{W}$ updates by $u_W/|W|$, which under this interpretation of $|W|^2$ as a moving average of $|u_W|^2$, means that $u_W/|W|$ corresponds to a RMSprop-like adaptive scaling of $u_W$. This allows us to view scale invariance as inducing an adaptive effect, and we construct the 2-Adam optimizer to explicitly capture this adaptive scaling, equivalent to applying Adam’s normalization procedure twice in succession.
> > >
> > > Thank you for your question, and please let us know if you have any additional questions regarding this aspect.
> > >
> > > > I don't understand this "η ̈φ = O(η2) + O(λη)," , and why you can drop this term, which hasn't mentioned in the main text.
> > >
> > > Thank you for your question! The term "η ̈φ = O(η²) + O(λη)" is neglected based on the fact that we assume both η and λ are set to small values. Specifically, η = 10⁻³ and λ = 10⁻² are the default values used in the AdamW implementation in PyTorch (https://pytorch.org/docs/stable/generated/torch.optim.AdamW.html). These small magnitudes allow us to reasonably disregard this second-order term, as we also empirically verify Figure 5.
> > > We have updated our derivation in Appendix J to clarify this point and to emphasize that our results are leading-order approximations that neglect higher-order terms. We greatly appreciate your feedback and believe the revision makes this aspect much clearer.
> > >
> > > **Summary:** We sincerely thank the reviewer for thoroughly examining our assumptions and derivations, as well as for suggesting a concrete set of experiments to empirically validate our theory. In response to your valuable feedback, we have motivated, clarified, and empirically verified the assumptions much more thoroughly in our paper, and conducted additional experiments using a larger transformer model on the suggested C4 dataset. We believe that these substantial updates have significantly improved the manuscript and addressed your concerns.
> > >
> > > In light of these improvements, we hope you might consider revising your score and supporting the acceptance of our paper. Please let us know if you have any additional questions or concerns.

---

> > ### Comment · Reviewer_bK6b · 2024-11-26
> > **re**
> >
> > i don't think you could say because $\eta=1e-3$ is small, we can drop the higher order term. first the scale of the weights are also small, with std  $ 1/width$; second the number of steps are large, usually $~1e5$.

---

> ### Author Response · Authors · 2024-11-29
>
> We thank the reviewer for increasing their score to 5 (though, as Reviewer GyMg comments, I don't think the official rating in your review has been updated?). We emphasize that the assumption that higher-order terms are small *relative* to leading-order terms is a standard assumption in the literature which has been used to uncover meaningful properties of practical optimization [1, 2, 3, 4]. For example, see Equation 3 of [4] (which was published as a spotlight paper at ICLR 2024). We argue that the overall contribution due to higher-order terms is still negligible *in comparison to* the overall contribution of leading-order terms. Importantly, the role of our empirical validation in Figure 1 and Figure 5 was to indeed verify that this assumption is reasonable and provides an accurate model of Adam.
>
> [1] Sadhika Malladi, Kaifeng Lyu, Abhishek Panigrahi, and Sanjeev Arora. On the sdes and scaling rules for adaptive gradient algorithms
>
> [2] André Belotto da Silva and Maxime Gazeau. A general system of differential equations to model first order adaptive algorithms
>
> [3] Anas Barakat and Pascal Bianchi. Convergence and dynamical behavior of the adam algorithm for non-convex stochastic optimization
>
> [4] Lizhang Chen, Bo Liu, Kaizhao Liang, and Qiang Liu. Lion Secretly Solves a Constrained Optimization: As Lyapunov Predicts

---

### Author Response · Authors · 2024-11-22
**Rebuttal: global response**

We thank the reviewers for their detailed comments, and are glad to hear that reviewers found our analysis to be “interesting and novel” (13ZQ) and to “maintain an interesting balance between theory and practical insight” via a “sound and grounded” theoretical study (GyMg).

We also appreciate the constructive feedback on clarifying the assumptions and derivations in Section 4 (bK6b, 13ZQ, GyMg), the suggestions for further scaling our experiments to validate our model (bK6b, 13ZQ, GyMg), and the pointers to related works to be discussed (GyMg, R4xU). To address these concerns, we have made the following substantial updates:

1. Increased the scale of our experiments:
    * Our language modeling experiments now use the **C4 dataset** and train with a transformer model of **30M parameters** (previously trained on a character-level shakespeare dataset using 10M parameters).
    * Provided **additional assessment of k-Adam** in this language modeling setting, robustly confirming **2-Adam to outperform Adam** in Appendix P.
    * Stronger empirical verification of our assumptions, now simulating the continuous-time trajectory in Figure 1 and 5 for 1,000 steps (corresponding to 100,000 continuous-time iterations), as well as verifying the assumptions of Section 4 in Appendix G for up to 1,000 steps (previously only 100 steps).
2. Structured our technical results and assumptions more clearly into lemmas and theorems throughout the paper.
3. Revised Section 4.2 and 4.3 to improve the presentation of our central findings relating to the **meta-adaptive effect of scale invariance** and the **2-Adam optimizer**, introducing supporting notation.
4. Provided a more comprehensive review of related literature in Section 5, and extended our discussion of how our findings relate to practice, the limitations of our work, and future research directions in Section 6.
5. Included a discussion on the special cases of RMSprop and signSGD in Appendix D.

We have highlighted our key modifications in red in our new manuscript.

The reviewers had many interesting questions and suggestions which we have addressed in our individual responses, supported by further additions to the main text and appendices. We hope that our newly updated manuscript better highlights our contributions towards understanding the theoretical properties of adaptive optimizers.

---

### Meta-Review · Area_Chair_qPzH · 2024-12-20

**Metareview:**

This paper introduces a continuous-time formulation of AdamW, deriving a stability region for its hyperparameters and exploring the role of normalization layers, leading to the proposal of k-Adam. While the approach offers a novel perspective on Adam's dynamics and suggests practical guidelines for hyperparameter selection, the theoretical framework relies on non-rigorous derivations and lacks strong experimental validation, particularly regarding the proposed k-Adam optimizer.  Given these weaknesses, the paper's contributions are not sufficiently robust to warrant acceptance in its current form.

**Additional Comments On Reviewer Discussion:**

The reviewers raise concerns about the rigor and clarity of the theoretical framework, particularly the assumptions and derivations in the continuous-time formulation of Adam(W). They also question the limited scale and diversity of the experimental results, especially regarding the newly proposed k-Adam optimizer.  The authors respond by increasing the scale of their experiments, providing stronger empirical verification of their assumptions, and clarifying the theoretical underpinnings and implications of their work. They also expand on the literature review, discuss the limitations of their approach, and address specific technical questions raised by the reviewers, including the motivation behind k-Adam and the relationship between stability and generalization.

The rebuttal did address some of the reviewers' concerns, but ultimately most reviewers do not feel like this paper is quite ready for publication.

---

### Decision · Program_Chairs · 2025-01-22

Reject